# Estimation of secondary organic aerosol formation parameters for the Volatility Basis Set combining thermodenuder, isothermal dilution and yield measurements

Petro Uruci[1,2], Dontavious Sippial[3], Anthoula Drosatou[1,2], and Spyros N. Pandis[1,2]

[1]Institute of Chemical Engineering Sciences (FORTH/ICE-HT), 26504, Patras, Greece
[2]Department of Chemical Engineering, University of Patras, 26500, Patras, Greece
[3]Department of Chemical Engineering, Carnegie Mellon University, Pittsburgh, USA

## Abstract

Secondary organic aerosol (SOA) is a major fraction of the total organic aerosol (OA) in the atmosphere. SOA is formed by the partitioning onto pre-existent particles of low vapor pressure products of the oxidation of volatile, intermediate volatility, and semivolatile organic compounds. Oxidation of the precursor molecules results in a myriad of organic products making the detailed analysis of smog chamber experiments difficult and the incorporation of the corresponding results into chemical transport models (CTMs) challenging. The volatility basis set (VBS) is a framework that has been designed to help bridge the gap between laboratory measurements and CTMs. The parametrization of SOA formation for the VBS has been traditionally based on fitting yield measurements of smog chamber experiments. To reduce the uncertainty of this approach, we developed an algorithm to estimate the SOA product volatility distribution, effective vaporization enthalpy, and effective accommodation coefficient combining SOA yield measurements with thermograms (from thermodenuders) and areograms (from isothermal dilution chambers) from different experiments and laboratories. The algorithm is evaluated with "pseudo-data" produced from the simulation of the corresponding processes assuming SOA with known properties and introducing experimental error. One of the novel features of our approach is that the proposed algorithm estimates the uncertainty of the predicted yields for different atmospheric conditions (temperature, SOA concentration levels, etc.). The uncertainty of these predicted yields is significantly smaller than that of the estimated volatility distributions for all conditions tested.

## 1. Introduction

Submicrometer atmospheric particles are of great importance due to their negative effects on public health (Pope and Dockery, 2006; Lim et al., 2012) and their uncertain

influence on Earth's climate (IPCC, 2021). Organic aerosol (OA) contributes 20–90 % to the submicron particulate mass (Zhang et al., 2007) and is emitted directly in the atmosphere as primary particles (POA) or formed as secondary organic aerosol (SOA). SOA constitutes a major fraction of the total OA in the atmosphere contributing more than 60 % on average (Kanakidou et al., 2005). SOA is formed by the condensation of low vapor pressure products of the oxidation of volatile (VOCs), intermediate volatility (IVOCs), and semi-volatile organic compounds (SVOCs).

Hundreds of mostly unknown products are formed during the oxidation of each SOA precursor making the detailed description of the corresponding reactions and eventual SOA formation extremely challenging. The volatility basis set (VBS) is one approach that has been proposed to simplify the system and to allow the SOA simulation in CTMs. The VBS describes the volatility distribution of OA using a set of surrogate species with effective saturation concentrations that vary by one order of magnitude (Donahue et al., 2006; Stanier et al., 2008). Volatility is one of the most important physical properties of SOA components as it determines to a large extent their gas-particle partitioning (Pankow, 1994a; 1994b). The parametrization of SOA formation for the VBS requires the determination of the yields of each volatility bin (volatility distribution of products) and the corresponding enthalpies of vaporization.

The SOA parametrizations for the VBS have been traditionally based on fitting yield measurements (Lane et al., 2008). The major weakness of this approach is that the resulting parametrization is limited to the range of OA concentrations and temperatures of the measurements. In most cases, the concentration range does not include the low concentrations relevant to the atmosphere and usually most of the experiments take place in a relatively narrow temperature range. Pathak et al. (2007a) needed 37 smog chamber experiments at different temperatures (0–45 $^{\circ}$C) and atmospherically relevant concentrations to constrain the α-pinene SOA temperature sensitivity.

A number of approaches has been used to minimize the number of experiments needed to characterize the temperature dependence of the SOA formation. Stanier et al. (2007) developed an experimental technique with which the temperature-controlled smog chamber could be heated or cooled after the SOA formation moving the system to new equilibrium favoring evaporation or condensation respectively. However, interactions of the SOA with the walls of the system increased the uncertainties of the approach. Stanier et al. (2008) presented an algorithm to fit the smog chamber

experiments using several volatility bins. However, the number of experiments needed by the algorithm should cover a wide range of concentrations and temperatures to effectively constrain the stoichiometric mass yields and the effective vaporization enthalpy.

In an effort to cover a wider concentration and temperature range, thermodenuder measurements can be used. The thermodenuder (TD) is a common instrument developed to characterize the volatility of atmospheric aerosols by heating them and observing the resulting changes in size, mass, optical properties, etc. (Burtscher et al., 2001; Wehner et al., 2002, 2004; An et al., 2007). TDs consist of a heated tube in which the volatile particle components evaporate followed by a cooling section with activated carbon to avoid vapor recondensation. The mass changes in TDs depend on the initial SOA concentration, the residence time in the heating tube, the vaporization enthalpy, and the mass transfer resistances. The latter are described by the effective accommodation coefficient that has been traditionally used to account for resistances to mass transfer not only at the surface of the particle but also inside the particle. The evaporation rate for most particles is relatively insensitive to its value when this value is around one. A typical way of reporting the TD measurements is by calculating the aerosol mass fraction remaining (MFR) at a given temperature after passing through the TD. The MFRs in a range of TD temperatures constitute the thermogram.

In applications in the field (Cappa and Jimenez, 2010; Huffman et al., 2009; Lee et al., 2010; Louvaris et al., 2017a) and in the laboratory (Kalberer et al., 2004; Baltensperger et al., 2005; An et al., 2007; Lee et al., 2011; Cain et al., 2020) the particles do not reach equilibrium with the gas phase inside the TD. Therefore, dynamic aerosol evaporation models (Riipinen et al., 2010; Cappa, 2010; Fuentes and McFiggans, 2012) are needed for the interpretation of TD measurements. Karnezi et al. (2014) used the time-dependent evaporation model of Riipinen et al. (2010) to calculate the OA volatility distribution, vaporization enthalpy, and mass accommodation coefficient from TD measurements. The authors showed that a simple error minimization approach may not be appropriate for such systems as very similar thermograms can be obtained for multiple combinations of different parameters. For this reason, their approach estimates an ensemble of "good" solutions, from which the best estimate and the corresponding uncertainties are derived.

Grieshop et al. (2009) suggested the combination of TD and isothermal dilution to constrain the volatility distribution of SOA. Karnezi et al. (2014) proposed an algorithm to include both types of measurements. The authors concluded that the combination of the two types of measurements can better constrain the OA volatility than each set separately. Louvaris et al. (2017b) and Cain et al. (2020) applied this algorithm to cooking OA (COA) and SOA, respectively. Louvaris et al. (2017b) showed that the use of only TD measurements led to overestimation of the SVOC fraction of COA, while the use of TD and isothermal dilution data reduces the uncertainty of the volatility distribution and the effective vaporization enthalpy. Cain et al. (2020) conducted TD and isothermal dilution experiments for α-pinene and cyclohexene ozonolysis SOA. The SOA in these two systems had similar thermograms, but different areograms. When only thermograms were used in the model, the volatility distributions were quite similar. However, the addition of areograms revealed that α-pinene ozonolysis SOA consists mostly of low-volatility organic compounds (LVOCs) and the cyclohexene ozonolysis SOA consists mostly of SVOCs.

To constrain the volatility product distribution of SOA and its effective vaporization enthalpy, we combine TD and isothermal dilution experiments with the SOA yield measurements. We extend here the algorithm of Karnezi et al. (2014) by introducing additional inputs (SOA yields) and by providing additional outputs (uncertainty of estimated yields in relevant atmospheric conditions). The algorithm is tested with "pseudo-experimental" data generated from the use of models simulating the corresponding measurement processes, therefore the true parameters are known. The results of the "pseudo-experiments" are corrupted so that they include experimental errors.

## 2. Model Description
### 2.1. SOA Formation

Gas-phase oxidation of VOCs involves a large number of reactions and produces a large number of products that can condense in the particulate phase. Depending on their effective saturation concentration, they can be represented in the 1D-VBS framework by

$$\text{VOC} + \text{oxidant} \rightarrow \alpha_1 P_1 + \alpha_2 P_2 + ... + \alpha_n P_n + \text{volatile products} \qquad (1)$$

where $n$ is the number of the surrogate compounds (volatility bins in the VBS), $P_i$ is the surrogate product in the $i$-th volatility bin and $\alpha_i$ is the corresponding stoichiometric mass yield. The total SOA mass yield can be then calculated as:

$$Y \equiv \frac{C_{OA}}{\Delta VOC} = \sum_i^n \frac{\alpha_i}{1+\left(C_i^*/C_{OA}\right)} \tag{2}$$

where $C_{OA}$ is the total SOA concentration, $\Delta VOC$ is the consumed concentration of the VOC and $C_i^*$ is the effective saturation concentration of compound $i$. This yield equation is an extension of the two-product model by Odum et al. (1996) replacing their semi-empirical partitioning coefficients with the assumption of a pseudo-ideal solution (Strader et al., 1999). This model assumes that the system has reached equilibrium when the yield was measured and that the differences in molecular weights are small.

The effective saturation concentrations at different temperatures are given by the Clausius-Clapeyron equation:

$$C_i^*(T) = C_i^*(T_{ref}) \frac{T_{ref}}{T} \exp\left[\frac{\Delta H_{vap,i}}{R}\left(\frac{1}{T_{ref}} - \frac{1}{T}\right)\right] \tag{3}$$

where $T_{ref}$ is the reference temperature in which the reference effective saturation concentration is defined (298 K in this work), and $\Delta H_{vap,i}$ is the enthalpy of vaporization of surrogate compound $i$.

## 2.2. Thermodenuder Model

The time-dependent evaporation of SOA in the TD used in this work is described by the dynamic mass transfer model of Riipinen et al. (2010). The evolution of the total particle mass, $m_p$, and the gas phase concentration of the compound $i$, $C_i$ are given by:

$$\frac{dm_p}{dt} = -\sum_{i=1}^n I_i \tag{4}$$

$$\frac{dC_i}{dt} = I_i N_{tot} \tag{5}$$

where $n$ is the number of surrogate compounds, $N_{tot}$ is the total number concentration of particles (assuming monodisperse aerosol population) and $I_i$ is the mass flux of compound $i$ from the gas to the particulate phase for each particle calculated by (Seinfeld and Pandis, 2016):

$$I_i = \frac{2\pi d_p M_i \beta_{mi} D_i}{R T_{TD}}\left(p_i - p_i^0\right) \tag{6}$$

where $d_\mathrm{p}$ is the particle diameter, $R$ is the ideal gas constant, $M_i$ is the molecular weight
of compound $i$, $D_i$ is the diffusion coefficient of compound $i$ in the gas phase at
temperature $T_\mathrm{TD}$, $p_i$ and $p_i^0$ are the partial vapor pressures of $i$ far away from the particle
and at particle surface, respectively, and $\beta_{\mathrm{m}i}$ is a factor for the correction of kinetic and
transition regime effects (Fuchs and Sutugin, 1970):
$$\beta_{\mathrm{m}i} = \frac{1 + Kn_i}{1 + \left(\dfrac{4}{3\alpha_{\mathrm{m}i}} + 0.377\right)Kn_i + \dfrac{4}{3\alpha_{\mathrm{m}i}}Kn_i} \tag{7}$$

where $Kn_i$ is the Knudsen number of compound $i$, and $\alpha_{\mathrm{m}i}$ is the mass accommodation
coefficient of compound $i$ on the particles. The partial vapor pressure of compound $i$ at
the particle surface is given by:
$$p_i^0 = x_{\mathrm{m}i} \frac{C_i^* RT}{M_i} \exp\left(\frac{4M_i\sigma}{RT_\mathrm{TD}\rho d_\mathrm{p}}\right) \tag{8}$$

where $x_{\mathrm{m}i}$ is the mass fraction of compound $i$ in the particulate phase, $C_i^*$ is the effective
saturation concentration, $\sigma$ is the surface tension (assumed 0.05 N m$^{-1}$ in our
simulations), $T_\mathrm{TD}$ is the particle temperature assumed to be the same as in the TD, and
$\rho$ is the particle density. The effective saturation concentrations at different TD
temperatures are given by Eq. (3).

Processes other than organic aerosol evaporation may affect the TD

measurements. For example, thermal decomposition may accelerate the transfer of
organic compounds from the particulate to the gas phase and may lead to overestimation
of the volatility of especially the least volatile components of the SOA (Epstein et al.,
2010; Saha and Grieshop, 2016; Stark et al., 2017). However, the corresponding
parameters for the SVOCs and the more volatile LVOCs that are important for
atmospheric SOA modeling should be a lot less uncertain given that they are measured
in relatively low TD temperatures. The use of isothermal dilution measurements may
also help identify cases in which the model does not include a process (e.g., thermal
decomposition) that dominates the behavior of the aerosol during heating. In this case,
one expects that the overall algorithm will have difficulties reproducing all
measurements (yields, isothermal dilution, and evaporation in the TD).

## 2.3. Isothermal Dilution Model

In isothermal dilution experiments, a SOA sample is injected in a reactor filled with clean air at room temperature. The concentrations of both the gas and particulate phase components are lowered due to dilution leading the system out of equilibrium. The evaporation of SOA as a result of isothermal dilution is also described by equations (3)-(8) (Karnezi et al., 2014), but the temperature is equal to 298 K. Evaporation in a dilution chamber depends on the initial SOA mass, time, and the $\alpha_m$, but not on $\Delta H_{vap}$ as the particles evaporate without a change in temperature.

The dilution ratio is an important parameter, varying typically from 10 to 20 in SOA experiments (Cain et al., 2020). Low dilution ratios result in little evaporation and little signal to be explored by the parameter estimation algorithm. High dilution ratios lead to very low initial concentrations in the dilution chamber and a lot of noise in the subsequent evaporation measurements.

## 3. Algorithm for the Estimation of VBS Parameters

The algorithm of Karnezi et al. (2014) was first extended to include an SOA partitioning model described by Equations (1) – (3) together with the TD and isothermal dilution models in order to estimate the volatility product distribution, vaporization enthalpy and accommodation coefficient. We discretized the domain of the parameters and simulated all combinations of stoichiometric mass yields ($\alpha_i$), $\Delta H_{vap}$, and $\alpha_m$. The yields $\alpha_i$ were allowed to vary from 0.0 to 0.8, with values of 0.0, 0.05, 0.1, 0.15, 0.2, 0.3, 0.4, 0.6, and 0.8. The user of the algorithm can specify an upper limit for the sum of the yields to reduce the number of the potential solutions that the algorithm will test. Combinations with sum of the yields exceeding 1.0 were excluded from the analysis originally. The sensitivity of our results to setting the upper limit of the sum of the yields equal to 2 is examined in Section 4.6. For a 4-product system there are 3,153 and for 6-product system 66,636 acceptable combinations. The values used for $\Delta H_{vap}$ were from 20 to 200 kJ mol$^{-1}$ with a step of 20, and for $\alpha_m$, the values used were 0.001, 0.01, 0.1, and 1. As a result 126,120 simulations are needed (computational time of about 15 h in an office PC) for a 4-product VBS and 2,665,440 for a 6-product solution.

For each simulation and each type of measurement, we calculated the *normalized mean square error* (NMSE) defined as

$$\text{NMSE} = \frac{\sum_{i=1}^{N_O} (P_i - O_i)^2}{\sum_{i=1}^{N_O} O_i} \qquad (9)$$


where $O_i$ represents the $i$th observed value (corresponding to a specific SOA
concentration for yield measurements or temperature for TD, or time for isothermal
dilution), $P_i$ the corresponding model-predicted value, and $N_O$ is the total number of
observations from each type of measurement. For each simulation (denoted as $s$), the
overall error was calculated by assuming equal weight to the set of yield, TD, and
dilution measurements and summing the corresponding errors:
$$E_s = \text{NMSE}_{Y,s} + \text{NMSE}_{\text{TD},s} + \text{NMSE}_{\text{Dil},s} \qquad (10)$$

The parameter combinations for which the overall error $E_s$ is less than 5% are

identified. The best solution is then calculated by averaging these solutions using the
inverse error $E_s$ as a weighting factor. The solutions that are closer to the measurements
have higher weight. Therefore, for every combination of $\alpha_i$, $\Delta H_{\text{vap}}$, and $\alpha_{\text{m}}$ the algorithm
calculates one overall *NMSE* following Eq. (10) and all data points for each solution
get the same weighting factor. More specifically the best estimate $\bar{x}$ is given by:
$$\bar{x} = \frac{\sum_{k}^{N} x_k \frac{1}{E_k}}{\sum_{k}^{N} \frac{1}{E_k}} \qquad (11)$$
where $x_k$ is the estimated value of a property (mass yield of a volatility bin, effective
vaporization enthalpy, or effective accommodation coefficient) and $N$ is the number of
combinations with error below the threshold value. The uncertainty range of the
parameters is estimated by calculating the standard deviation ($\sigma$):
$$\sigma = \sqrt{\frac{\sum_{k}^{N} \left[ (x_k - \bar{x})^2 \cdot \frac{1}{E_k} \right]}{\sum_{k}^{N} \frac{1}{E_k}}} \qquad (12)$$
following Karnezi et al. (2014).

## 4. Testing of the Algorithm

### 4.1. Generation of Data for Evaluation

In order to evaluate the algorithm, we generated data using the output of SOA formation, thermodenuder and isothermal dilution models described in Section 2 for systems with known volatility distribution of the products, and properties. Then, these data were "corrupted" with random errors to represent the "noise" observed in laboratory measurements for yields, thermograms, and areograms. As a result, there is no set of model parameters that can reproduce all the "measurements". The yields were corrupted based on the variability of laboratory measurements of Pathak et al. (2007a), by assuming a normal distribution and standard deviation ($\sigma_Y$) given by:

$$\sigma_Y = 0.1 Y_{\text{true}} + 0.02 \tag{13}$$

where $Y_{\text{true}}$ are the correct yields.

For TD, the errors were calculated by assuming a normal distribution and the standard deviation ($\sigma_{\text{TD}}$) suggested by Karnezi et al. (2014):

$$\sigma_{\text{TD}} = 0.51 MFR_{\text{TD,true}} - 0.5 \left( MFR_{\text{TD,true}} \right)^2 \tag{14}$$

where $MFR_{\text{TD,true}}$ are the correct MFR values for each TD temperature.

For dilution, the errors were calculated by assuming a uniform distribution and standard deviation ($\sigma_{\text{Dil}}$) suggested by Karnezi et al. (2014):

$$\sigma_{\text{Dil}} = 0.05 MFR_{\text{Dil,true}} + 0.03 \tag{15}$$

where $MFR_{\text{Dil,true}}$ are the correct MFR values for isothermal dilution.

Based on the above methodology, we generated "pseudo-measurements" of yield, TD, and isothermal dilution for different SOA systems. The parameters used to produce the pseudo-experimental data are summarized in Table S1. The "experimental" conditions assumed for the TD and isothermal dilution measurements are shown in Table S2.

In "Experiment" A, we test the performance of the algorithm against α-pinene ozonolysis data and examine the effect of TD and isothermal dilution data. For "Experiment" A, the "true" values were taken from the parameterization derived by Pathak et al. (2007b) for the ozonolysis of α-pinene at low $NO_x$, dark and low RH conditions. Therefore, these results are good fits of the measurements analyzed in that study. The parametrization was derived assuming a 4-volatility bin system with saturation concentrations ranging from 1 to $10^3$ μg m$^{-3}$. The effective vaporization enthalpy estimated in that study was equal to 30 kJ mol$^{-1}$. Because the effective

accommodation coefficient was not part of the Pathak et al. (2007b) parametrization, we assumed a value of 0.5 in this work. We used a small number of yield measurements at atmospherically relevant SOA concentrations of 1, 5, 10, 20 and 40 $\mu g\ m^{-3}$ (Fig. 1). For this SOA system, the yield at 40 $\mu g\ m^{-3}$ did not exceed 20%. The thermogram includes ten MFR data points in the temperature range of 20 to 200 $^{o}$C. For the highest temperature, more than 70% of the SOA mass was evaporated. The areogram shows that the correspondent SOA evaporated almost by 70 % in the first 0.5 h and more than 90% in less than 3 h.

For "Experiment" B, the "true" values were taken from the alternative parametrization proposed by Pathak et al. (2007b) for the same oxidation system as described before. This time, the authors used a 7-volatility bin system with saturation concentrations ranging from $10^{-2}$ to $10^{4}$ $\mu g\ m^{-3}$ in their parametrization. The effective vaporization enthalpy of the parametrization was 30 kJ $mol^{-1}$, while for the accommodation coefficient we assumed again a value of 0.5. The yield, TD and isothermal dilution "measurements" of Experiment B are generated in the same SOA mass concentration, temperature, and dilution time range as in the previous pseudo-experiment (Fig. 2).

For "Experiment" C, the "true" values were based on the parameterization of the SOA formed during α-humulene ozonolysis by Sippial et al. (2022). The authors measured high SOA yields for α-humulene in the main smog chamber (~70% at 60 $\mu g$ $m^{-3}$), and their corresponding thermogram suggested that the SOA particles fully evaporated at 150 $^{o}$C, while the areogram showed modest (20%) evaporation in the dilution chamber after 3 hours. A 4-volatility bin set with saturation concentrations ranging from $10^{-2}$ to 10 $\mu g\ m^{-3}$ was used in that study to fit the measurements. The stoichiometric coefficients of the three least volatile bins ($10^{-2}$, $10^{-1}$ and 1 $\mu g\ m^{-3}$) were around 0.1 and for the most volatile (10 $\mu g\ m^{-3}$) 0.25. The vaporization enthalpy was 115 kJ $mol^{-1}$ and the accommodation coefficient was 0.01 (Table S1). We assumed five yield "measurements" in the SOA concentration range of 1 to 100 $\mu g\ m^{-3}$ with yield values as high as 65 % at 100 $\mu g\ m^{-3}$ (Fig. 3). The corresponding thermogram consisted of 10 data and the particles fully evaporated at TD temperatures higher than 150 $^{o}$C. The areogram consisted of 17 data points and only 20 % of the SOA evaporated in the dilution chamber.

## 4.2. Parameter Estimation for "Experiments" A, B, and C

We explored the performance of the algorithm for different choices of the number of volatility bins, the range of saturation concentrations, and the range of SOA mass concentration range in the yield measurements. For each test, the "true" and the estimated properties are summarized in Table 1.

We evaluated the performance of our parameter estimation algorithm comparing its predictions both against the "measurements" and the "truth" defined as the predictions of the original parameterization. In both comparisons, *mean normalized error* (MNE) (Emery et al., 2017) was used as the evaluation metric because it has a simpler physical meaning than NMSE.

For the evaluation against the "measurements", the $MNE_M$ was defined as

$$MNE_M = \frac{100}{N_O} \sum_{i=1}^{N_O} \frac{|EST_i - O_i|}{O_i} \tag{16}$$

where $EST_i$ is the estimated by the algorithm value and corresponds to a specific measured point $O_i$.

For the evaluation against the "truth", which includes conditions (e.g., temperatures or concentrations) for which there are no available measurements, the $MNE_T$ was defined as:

$$MNE_T = \frac{100}{N_d} \sum_{j=1}^{N_d} \frac{|EST_j - TR_j|}{TR_j} \tag{17}$$

where $EST$ and $TR$ are the estimated and the "true" values respectively. $N_d$ is the total number of data points included in calculations and depends on the selected discretization of the corresponding dependent variable (e.g., SOA concentration, TD temperature, and dilution time). We used a linear discretization for the SOA concentrations (from 0.01 to 50 μg m$^{-3}$ with a step of 0.01) and the TD temperatures (20 to 200 °C with a step of 5 °C but excluding zero MFR values to avoid the division by zero). For the dilution time, the sampling time step was not constant. We used a higher resolution for the first 0.5 hour (step of 2 min), in which the evaporation is usually faster, and a lower resolution afterwards (step of 10 min).

Finally, we used the average relative standard deviation (*ARSD*) as a metric to quantify the uncertainty of the estimates (range of good solutions) using the same discretization as in the $MNE_T$ metric. The *ARSD* is given by:

$$ARSD = \frac{100}{N_d} \sum_{j=1}^{N_d} \frac{\sigma_j}{EST_j} \tag{18}$$

where $\sigma_j$ is the standard deviation for data point $j$.

### 4.2.1 Parameter Estimation for "Experiment" A

In Test A1, we applied the algorithm in the same range of saturation concentrations and with the same number of volatility bins as these used to produce the "experimental" data. The upper bin ($10^3$ µg m$^{-3}$) exceeded the maximum SOA concentration (40 µg m$^{-3}$) in the measurement range by one order of magnitude.

Figure 1 depicts the estimated and the range of the ensemble of best solutions for the three types of "measurements" for Test A1. There were 148 "good" solutions under the 5% threshold out of the 126,120 simulations (Table S3). The density distribution of the solutions is depicted in Figure S1. The performance of the model for the yields at 25 $^{\circ}$C was quite encouraging with a small tendency of overprediction for SOA higher than 10 µg m$^{-3}$. The $MNE_M$ of the model for the SOA yield "measurements" (given by Eq. 16) was equal to 25% (Table 2). The corresponding discrepancy between the true parameterization and the measurements (due to the measurement error that we introduced) was 21.2% (Table 2). This indicates that a significant part of the algorithm error can be explained by the uncertainty introduced in the measurements.

Our algorithm can be used to calculate the SOA yield at different concentrations and temperatures. The yields were calculated in the atmospherically relevant range of 0–50 µg m$^{-3}$ SOA concentration and at four temperatures (5, 15, 25, and 35 $^{\circ}$C) using the true parameter values and the estimated parameters of Test A1 (Fig. 1a-d). At 25 $^{\circ}$C (Fig. 1c), the estimated yield curve is in good agreement with the "true" yield curve for SOA concentrations lower than 6 µg m$^{-3}$ (error of 8% at 6 µg m$^{-3}$), but the discrepancies increase at higher concentrations (error of 23% at 50 µg m$^{-3}$). The average $MNE_T$ error between the true parametrization and the estimated values (given by Eq. 17) was equal to 17.3% for yields at 25 $^{\circ}$C (Table 3). The uncertainties, as expected, are larger at lower temperatures. However, the $MNE_T$ error (estimated yields compared to the true value) remains less than 25% (Table 3) even at 5 $^{\circ}$C, quite far from the measurement temperature. Both $MNE_T$ and $MNE_M$ were quite close to the introduced experimental error. Their difference can be explained by both the "noise" introduced to the

"measurements" that affects $MNE_M$ and the higher number of points used to calculate
$MNE_T$.

The SOA model used in this work assumes that the stoichiometric coefficients

($\alpha_i$) are temperature independent. Therefore, processes, such as formation of highly
oxygenated organic molecules (HOMs) and oligomerization which are expected to be
temperature dependent (Quéléver et al., 2019; Gao et al., 2022), are not described by
our algorithm.

The algorithm provides a range of "good" estimates in addition to the best

estimate. The range can be defined by the lower and upper SOA yield limits of the
ensemble of the good solutions at each point. At 25 °C, the yield range increased, as
expected, at higher concentrations (yield range of 0.05 at 1 μg m$^{-3}$ to 0.17 at 50 μg
m$^{-3}$). The average relative standard deviation ($ARSD$ of the estimated yields defined by
Eq. 18) was equal to 26% (Table 4) for the 25 °C case. For the rest of the temperatures,
the $ARSD$ increased for the lower temperatures, ranging from 24% at 35 °C to 35% at
5 °C (Table 4) and including in all cases the true solution.

For the TD (Fig. 1e), the model reproduced well the correspondent thermogram

with low errors compared to the "measurements" with an error $MNE_M$ of 7% (Table 2).
The error $MNE_T$ compared to the "true" values was 5.5% (Table 3). The error of the
TD "measurements" compared to the true values was equal to 7.6% (Table 2).
Therefore, the error of the proposed algorithm is quite similar to the experimental error.
The error introduced into the "measurements" was transferred, as expected, to the error
metrics of the algorithm.

For the isothermal dilution (Fig. 1f), the algorithm did reasonably well for the

first 30 minutes and then the evaporation was slightly underpredicted leading to an error
$MNE_M$ of 16.7% (Table 2). This $MNE_M$ value was roughly two times higher than the
corresponding error between the dilution measurements and the true parametrization
(Table 2). The error between the estimated and the "true" values $MNE_T$ was 19%. The
$ARSD$ of 24% (Table 4) was sufficient to include the true solution.

The estimated volatility distribution of the products and the effective

vaporization enthalpy and accommodation coefficient using the three types of
measurements can be seen in Figure 4 and Table 1. The estimated volatility distribution
of the products was in a good agreement with the "true" values ($\alpha_i$ absolute difference
of 0.01 at 1 μg m$^{-3}$, 0.03 at 10 μg m$^{-3}$, 0.07 at $10^2$ μg m$^{-3}$, and 0.04 at $10^3$ μg m$^{-3}$) and
the estimated uncertainties contained the correct values. There is a large uncertainty

range for the two higher volatility bins (standard deviation higher than 0.13) indicating that yield values at higher SOA concentrations would be needed to better constrain these volatility bins. The relative error of the estimated $\Delta H_{vap}$ is 10%. The estimated accommodation coefficient was 0.17 compared to a true value of 0.5. The estimated uncertainty for the effective accommodation was almost one order of magnitude (from 0.06 to 0.51) indicating the difficulty of constraining this parameter when it is close to unity and thus the resistances to mass transfer are small.

**4.2.2 Parameter Estimation for "Experiment" B**

In this section, we analyze the pseudo-experimental data of Experiment B, which were obtained from the parametrization of the same smog chamber results used in Experiment A, but with more components and a much wider range of volatilities including LVOCs, SVOCs and IVOCs ($10^{-2}$–$10^4$ µg m$^{-3}$). In Test B1, the algorithm was applied using a 4-bin VBS with saturation concentrations ranging from 1 to $10^3$ µg m$^{-3}$. In this test, we attempted to model the behavior of the system with a narrower volatility range than the real one. The upper limit of the saturation concentration range that we tested did not exceed the $10^3$ µg m$^{-3}$ because Experiment B took place in moderate SOA concentration levels (up to 40 µg m$^{-3}$), which means that it is practically impossible to constrain the $10^4$ µg m$^{-3}$ or higher volatile bins. Figure 2 shows the results of the fitting for the three types of "measurements" in this experiment. There were 82 "good" solutions under the 5% threshold out of 126,120 simulations (Table S3) and the density of the solutions are shown in Figure S2. At 25 °C, the model performance for the yields is encouraging ($MNE_M$=20.6%). This is again pretty close to the measurement error (20.5%). By comparing the estimated and the "true" yield curves at 25 °C, the error $MNE_T$ is now 14%. The error increases to 31% at 5 °C, far from the available measurements. This is reflected also in the increase of the uncertainty of our estimates with the $ARSD$ increasing from 17% at 35 °C to 37% at 5 °C (Table 4). Once more the uncertainty range estimated by the algorithm includes the true values.

Both "measured" and "true" thermogram were well captured by the best estimate ($MNE_M$ of 6% and $MNE_T$ of 4%) with an uncertainty $ARSD$ of 20.5%. The evaporation in the dilution chamber was a little underestimated for the first 2 h, but then it was slightly overpredicted. The $MNE_T$ for the areogram was 13.3% and the true values were included within the range of the estimates ($ARSD$ of 18%).

Figure 5 shows the results of Test B1 for the volatility distribution of the

products. The "true" stoichiometric coefficient for the 1 μg m$^{-3}$ bin was overestimated
by 0.01 by the algorithm. This overestimation actually corresponds to the total material
of the 10$^{-2}$ and 10$^{-1}$ μg m$^{-3}$ bins of the "true" system. This indicates that the algorithm
places the material of the two lowest bins that are not part of the solution to the bin with
the lower volatility. For the 10 μg m$^{-3}$ and 10$^{2}$ μg m$^{-3}$ bins, the relative errors between
the estimated and "true" were 58% and 277% respectively (Table S4), while for the 10$^{3}$
μg m$^{-3}$ bin, the relative error was 10 %. The $\Delta H_{vap}$ was predicted accurately (error of
only 4%), while $\alpha_m$ was underpredicted (0.1 instead of 0.5). The model compensates
for the missing volatility bins by increasing the material in the 10$^{2}$ μg m$^{-3}$ bin and by
decreasing the accommodation coefficient.

The results of Test B1 suggest that the mismatch between the actual SOA

volatility distribution and the range used for the fits can introduce significant errors in
the retrieved distribution for individual volatility bins. However, despite these
problems, the yields predicted by the derived parameterizations have a much lower
error than the volatility distribution. This is a valuable insight for the strengths and
weaknesses of this and other similar SOA parameter estimation algorithms.

**4.2.3 Parameter Estimation for "Experiment" C**
In Test C1, we obtained the best fits for the pseudo-measurements of Experiment C by
applying the algorithm in the same range of saturation concentrations and with the same
number of volatility bins (4 volatility bins in the 10$^{-2}$–10$^{1}$ μg m$^{-3}$ saturation
concentration range) as the true volatility distribution.

Figure 3 shows the results of the fitting for the three types of "measurements".

There were 3,479 "good" solutions under the 5% threshold out of the 126,120
simulations (Table S3). The density distribution of the solutions is shown in Figure S3.
The best estimate for the SOA yields at 25 $^{\circ}$C was in a good agreement with the
"measurements" ($MNE_M$=6.3%) and the "true" values ($MNE_T$=9.6%). For the rest of
the temperatures, there was a decreasing trend of the error as the temperature decreased
varying from 15.5% at 35 $^{\circ}$C to 6.2% at 5 $^{\circ}$C. A similar decreasing trend was observed
for the uncertainty $ARSD$ of the estimates which varied from 23% at 35 $^{\circ}$C to 15% at 5
$^{\circ}$C. This behavior is the opposite from what we observed in the previous tests, in which
both errors and uncertainties increased at lower temperatures. However, the changes in
both the error and the uncertainty are small (change of around 7% between the upper
and lower temperature for both metrics), indicating that this system is less temperature-
sensitive in this temperature range than the previous ones.

The performance of the algorithm was satisfactory compared to the TD

"measurements" ($MNE_M$=12.9%). The corresponding error of the algorithm for the true
values ($MNE_T$) was 4.4% for temperatures up to 110 $^o$C and equal to 10.6% for the
lower values at higher temperatures. According to Figure 3, the evaporation due to
dilution was initially overestimated for the first 30 min, but then underestimated
(highest MFR discrepancy of 0.05) and there is a high uncertainty range of the
corresponding estimates (MFR range of 0.46 at 3 h). However, the low dilution values
resulted in low relative errors ($MNE_M$ of 3.5% and $MNE_T$ of 2.7%).

Figure 6 shows that the highest relative errors were calculated for the $10^{-1}$ and

$10^0$ μg m$^{-3}$ bins (23% and 33% respectively), and smaller relative errors for the other
two bins (less than 13%). The uncertainties were almost of the same magnitude for all
bins with standard deviations ranging from 0.09 to 0.13. The performance of the model
was good for the $\Delta H_{vap}$ (relative error of 7%), but with high uncertainty for $\alpha_m$.

**4.3. Effect of the Volatility Range**

In in this section, we explore the performance of the algorithm for different choices of
the number of volatility bins and the range of saturation concentrations. The analysis
of the results of Test B1 has already quantified the effects of using a narrower volatility
distribution in the parameter estimation algorithm than the one of the investigated SOA
system. Additional sensitivity tests are performed here for all cases.

In Test A2, we used 3 volatility bins covering the $1-10^2$ μg m$^{-3}$ saturation

concentration range instead of the 4 bins used in Test A1. The narrower assumed
volatility range had a very small effect on the estimated yields at all temperatures (Table
3 and Fig. S4) compared to Test A1. The change in $MNE_T$ ranged from 3% at 5 $^o$C to
0.3% at 35 $^o$C. Minor changes were detected in the predicted thermogram (change of
0.8%) and areogram (change of 0.5%) as well. The uncertainty of the yield estimates
increased by less than 2.5% at all temperatures. The estimated volatility distribution of
the SOA products of Test A2 changed by less than 5% in the two lower bins. The
material in the $10^2$ μg m$^{-3}$ increased by 15% to account for the SOA of higher volatility
that could not be included otherwise in the estimated distribution. The estimated $\Delta H_{vap}$
was in this case 32 kJ mol$^{-1}$ (2.7% decrease) and the $\alpha_m$ decreased by 12% with respect
to Test A1.

In Test A3, we shifted the assumed 4-bin volatility distribution by one order of magnitude to lower values (from 1–1000 $\mu$g m$^{-3}$ in Test A1 to 0.1–100 $\mu$g m$^{-3}$ in Test A3). In this case, the algorithm distributed exactly the same material to the 1, 10 and 100 $\mu$g m$^{-3}$ volatility bins as in Test A2, and it predicted correctly zero SOA in the 0.1 $\mu$g m$^{-3}$ bin (Table 1). The $\Delta H_{vap}$ and $\alpha_m$ estimated values were also unchanged with respect to Test A2. This in turn, led to the same estimated yields at different temperatures (no change in the error between the two tests).

In Test C2, we applied the algorithm against the Experiment C "measurements" using a 4-volatility bin system in the 1 to $10^3$ $\mu$g m$^{-3}$ range, that is two orders of magnitude higher than the actual range of the "true" values. Figure 7 shows the results of the fitting for the three types of "measurements". Despite the significant mismatch of the volatility distributions the $MNE_M$ increased by only 2.3% for the estimated SOA yields. The error for the TD measurements increased by 20% while it actually decreased a little (1.2%) for the dilution data. The errors compared to the true values increased by less than 3% for the temperature range 15–35 $^o$C while it increased by 12% at 5 $^o$C. These results suggest that the estimated yields are quite robust in this case to the assumed volatility range. The major effect of the mismatch in volatility ranges was evident in the predicted thermogram with overestimation of the MFR for the 60–120 $^o$C temperature range and underprediction in higher temperatures. The increase in $MNE_T$ for the TD MFR was 17.2% (Table 3). The change in the predicted areogram was marginal and led to a small increase of $MNE_T$ (error increase by 0.7%) (Table 3). The algorithm underestimated again the $\alpha_m$ (0.004 instead of 0.01) but also recognized the high uncertainty of the corresponding estimate. The algorithm distributed significant material to the 1 $\mu$g m$^{-3}$ bin (3.6 times higher than the actual), in an effort to account for the absence of the $10^{-2}$ and $10^{-1}$ $\mu$g m$^{-3}$ bins. The $\Delta H_{vap}$ was underestimated with an error of 21%.

The results of the above tests indicate that a mismatch between the true and assumed volatility ranges of the SOA increases in general the estimation error but the increase is small to modest. This is reassuring for the robustness of the proposed algorithm.

## 4.4. Effect of Measurements at High SOA Levels

During the last decade there has been a significant shift of the performed SOA smog chamber towards lower SOA concentrations. This is needed to increase the accuracy at

the ambient concentration levels. The high SOA concentration experiments that once represented the majority of the performed experiments are becoming increasingly rare. In this paragraph we examine the value of these high concentration experiments for the estimation of SOA yields at ambient conditions.

To examine the effect of "measurements" at SOA levels much higher than the atmospheric ones, we included an extra yield measurement at 200 μg m$^{-3}$ in the yield data of Experiments A and B. In Test A4 and B2, we applied the algorithm once again against the three types of "measurements" by using a 4-volatility bin system with saturation concentrations ranging from 1 to $10^3$ μg m$^{-3}$.

In Test A4, the additional experiment at high SOA concentration led to an $MNE_T$ of 15.7% for the yields at 25 $^o$C (Table 3 and Fig. S5), which is by 1.6% lower than that without this experiment in Test A1. The improvement was more significant at lower temperatures e.g., the $MNE_T$ at 5 $^o$C was reduced from 24.4% to 20.4%. The reduction in the $ARSD$ for the SOA yields ranged from 3.8% at 5 $^o$C to 0.9% at 35 $^o$C (Table 4). Figure 8 depicts the results of the model for the yields and the volatility distribution of the products for Test A4. The accuracy of the predicted volatility distribution increased especially for the higher volatility material. For example, the error for the $10^2$ μg m$^{-3}$ bin was reduced from 41% in Test A1 to 6% in this case (Table S3). Minor changes in the errors were detected for the $\Delta H_{vap}$ and $\alpha_m$ between the two tests (3% increase and 6% decrease respectively).

Similar to Test A4, in Test B2 we added a yield measurement at 200 μg m$^{-3}$ in the Experiment B set of "measurements". Figure 9 depicts the results of the model for the SOA yields at 25 $^o$C and the estimated volatility distribution of the products. The use of the additional data point led to a reduction of the $NME_T$ from 13.9% in Test B1 to 9% in Test B2 at 25 $^o$C (Table 3). Similar reductions in the $NME_T$ were observed for the other temperatures, with the highest one observed at 5 $^o$C (lower error by 7%) (Figure 10). The reduction in the $ARSD$ for the estimated yields ranged from 3.3% at 5 $^o$C to 1.2% at 35 $^o$C (Table 4). Minor changes were observed for the estimated thermogram (Fig. S6) (change in the $NME_T$ of 1.5%) and the uncertainty of the estimates (change in the $ARSD$ of 2.5%). The error in the estimated areogram was also small but in this case the error increased by 5%. The additional data point helped decrease the errors for the estimated mass of the more volatile SOA products (Fig. 9) and especially for the $10^2$ μg m$^{-3}$ bin. The $\Delta H_{vap}$ and $\alpha_m$ estimated values were only slightly affected by the additional measurement.

By comparing the results Tests B1 and B2 with Case A, one would expect that

the retrieved volatility distribution of the products will be quite similar. The differences
present are due to a large extent to the different random experimental errors introduced
in the two sets of "measurements" for Experiments A and B. A second reason for the
differences is that parametrizations of the two "Experiments" by Pathak et al. (2007b)
even if they were derived from the same smog chamber experiments have some
differences. As a result, the "true" yields, thermogram, and areogram in Cases A and B
are not exactly the same (Figs. 1 and 2).

These results suggest that an additional yield measurement at high SOA can

lead to a substantial reduction of the error for the estimated yields at low temperatures
(Fig. 10) and also a better estimation of the SOA products with higher volatility ($10^2$
and $10^3$ µg m$^{-3}$). These products may contribute little to the SOA concentration at 25
$^{\circ}$C, but their reactions (aging) could lead to significant additional SOA in later stages.

**4.5. Significance of Each Type of Measurement for the Parametrization**

To quantify the effect of each type of measurement for the parameter estimation and
their subsequent effect on the estimated SOA yields, we repeated tests A1, B1, and C1
withholding one set of measurements. More specifically, we used the algorithm
providing it the following combination of measurements: TD and isothermal dilution,
SOA yields and isothermal dilution, and finally SOA yields and TD.

The use of only the TD and isothermal dilution data corresponds for all practical

purposes to the previous algorithm of Karnezi et al. (2014) which has been the starting
point of this work. In Test A1, the absence of the yield measurements led to a significant
deterioration of the ability of the algorithm to estimate SOA yields at all temperatures
and concentrations (Fig. S7). The SOA yield error of the algorithm in the 5–35 $^{\circ}$C
temperature range increased from 14-24% (when all measurements are provided) to
approximately 100% (Table S5). The corresponding uncertainty range also increased
by a factor of 4-6 (Table S6). Similar results were obtained in the other tests.

Figure S8 shows the volatility distribution of the products, $\Delta H_{vap}$ and $\alpha_m$ in Test

A1. High discrepancies and uncertainties were observed for the estimated
stoichiometric coefficients ($\alpha_i$), with an increase in the relative error by a factor of 3-4
for the 1 and 10 µg m$^{-3}$ bins (Table S7) compared to the case when all three types of
measurements are used.

Figures S9 and S10 show the results of the algorithm for Test A1 when only the

SOA yields and isothermal dilution measurements are provided as inputs to the
algorithm. In this case the algorithm cannot constrain well the $\Delta H_{vap}$ (relative error of
almost 270% with respect to the true value) as a result of the missing TD measurements.
This led to significant increase in the $MNE_T$ for the estimated yields when moving far
from the temperature of the measurements (MNE of 65% at 15 $^{o}$C and 122% at 5 $^{o}$C).

Figures S11 and S12 show the results of the algorithm for Test A1 when only

yield and TD measurements are provided as inputs. In this case, there was a significant
reduction in the error for the $\Delta H_{vap}$ respect to the previous case (from 270% to 50%),
but it was still much higher than the 10% error when all three types of measurements
were used. This led to better agreement between the true and estimated yields at lower
temperatures ($MNE_T$ of 23% and $ARSD$ of 44%).

When comparing TD/Dilution, Yields/Dilution, and Yields/TD results, the

Yield/TD combination gave the best results out of the three pairs. The isothermal
dilution measurements are the least valuable of the three because only a relatively small
fraction of the SOA evaporates and therefore the information provided is relatively
limited and focuses on the more volatile components of the particles. Also, TD
measurements are important to constrain well $\Delta H_{vap}$ and allow the more accurate
extrapolation of the results to other temperatures. also provides information for the
volatility distribution of the OA. However, our results suggest that the combination of
the three types of measurements does need to a major improvement over either the
TD/Dilution approach or the Yield/TD approach.

### 4.6. Sensitivity to the Upper Limit of the Sum of Product Yields

The maximum sum of the VBS product yields is one of the parameters that the user of
the algorithm chooses. In the analysis so far a value of 1 had been selected to reduce
the computational cost of the algorithm. Selected tests were repeated using a maximum
sum of 2 to quantify the effects of this choice on the estimated parameters and more
importantly on the SOA yields predicted by the parameterization. For a 4-product
system there are 9,191 product yield combinations and considering the discretization of
$\Delta H_{vap}$ and $\alpha_m$, this leads to a total of 367,120 simulations (Table S3).

The increase in the upper limit of the sum of the yields led to an increase in the

"good" solutions in Tests A1, A4, B1, B2, and C2. The additional solutions had
different yields mostly in the $10^3$ µg m$^{-3}$ bin. This led to an increase of the mass yield

of this bin by 37% in Test A1, 47% in Test B1, and 29% in Test C2 (Table S8). The uncertainties were even higher showing once again the difficulty to constrain the IVOC range where there are no SOA measurements at very high SOA concentrations. The new parametrizations had a minor effect on the estimated yields at different temperatures with maximum change in the $MNE_T$ found at 5 $^{\circ}$C (change of 1.8% in Test A1 and 1.2% in Test B2) and much smaller otherwise (Table S9). Therefore, the use of the higher upper limit has an effect on the estimate of the $10^3$ μg m$^{-3}$ bin which is quite uncertain in all cases, but has a minor effect on the predicted SOA yields at ambient conditions.

**5. Conclusions**

An algorithm was developed to estimate VBS parameters for SOA formation combining yield measurements from atmospheric simulation chambers with thermodenuder and isothermal dilution measurements chambers. An additional feature of this approach is that the algorithm estimates the uncertainty of the predicted SOA yields for different SOA concentrations and temperatures, assisting in this way in the design of future experiments.

The algorithm was evaluated against pseudo-experimental data for SOA systems with known properties. The algorithm performed quite well at reproducing the SOA yields at atmospherically relevant concentrations and temperatures with errors less than 20% for practically all cases. This was the case even at temperatures as low as 5 $^{\circ}$C and also when the volatility range used for the parameter estimation was narrower than that of the simulated SOA system. One should note that this error was quite similar in most cases to the experimental error assumed in the construction of the "measurement" datasets.

The errors in the retrieved SOA volatility distributions were in general higher than those of the SOA yields. This is due to a large extent to the existence of multiple solutions that can result in similar yields. The accuracy of the estimated mass fractions of the more volatile SOA components improved when an additional yield measurement at high SOA (e.g., at 200 μg m$^{-3}$). The addition of this measurement also improved the estimated yields at low temperatures. This therefore suggests that data points at high SOA concentrations should also be obtained experimentally, together with the data points at atmospherically relevant atmospheric SOA levels.

In all cases the algorithm results in good estimates of the effective evaporation
enthalpy. On the other hand, the estimates of the effective accommodation coefficient
are usually quite uncertain. The effect of the mass accommodation coefficient on the
measured quantities is relatively small compared to the other parameters (volatility
distribution, effective evaporation enthalpy) making it difficult to constrain. This
conclusion is consistent with the results of Karnezi et al. (2021). The addition of the
SOA yields to the inputs does not make much of a difference, because these are not
affected by the accommodation coefficient.
The approach combining yield, TD (thermograms), and isothermal dilution
(areograms) measurements is recommended for future parametrizations of SOA
formation. The use of the results of these experiments that have been designed for the
measurement of SOA yields to other applications (e.g., new particle formation) should
be performed with caution. Our results indicate that the derived parameterizations are
able to predict the SOA yields under different atmospheric conditions with errors of
around 20% or less, but the derived volatility distributions can be quite uncertain. These
uncertainties are higher for the tails of the distribution (the low volatility and the
intermediate volatility organic compounds). Different experiments should be probably
performed for the derivation of the VBS distribution if for example on is interested in
new particle formation and therefore the low volatility organics focusing on low SOA
concentration levels and the least volatile SOA components.


**6. Code and data availability**
The    code    and    simulation    results    are    available    upon    request
(spyros@chemeng.upatras.gr).

**7. Supplementary information**

**8. Author contribution**
PU, and SNP designed the research. PU developed the final model code. AD developed
a first version of the code and performed preliminary feasibility tests. DS and SNP
designed the experiments for the α-humulene ozonolysis and DS carried them out. PU
performed the simulations, the formal analysis, and wrote the original draft. Paper
review and editing was performed by SNP.

## 9. Competing interests

The authors declare that they have no conflict of interest.



## 10. Financial support

This work has been supported by the Chemical evolution of gas and particulate-phase organic pollutants in the atmosphere (CHEVOPIN) Project of the Hellenic Foundation for Research and Innovation (HFRI) under grant agreement no. 1819 and the European Union's Horizon 2020 research and innovation program through the EUROCHAMP-2020 Infrastructure Activity under grant agreement no. 730997.

718

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

**Table 1:** True and estimated volatility distribution of the products for 8 different tests. The uncertainty of the estimates (±σ) is also included.

| TEST | $\Delta H_{vap}$ (kJ mol$^{-1}$) | $log(\alpha_m)$ | Stoichiometric Coefficients ($\alpha_i$) at $C_i^*$ (µg m$^{-3}$) | | | | | | |
|---|---|---|---|---|---|---|---|---|---|
| | | | $10^{-2}$ | $10^{-1}$ | $10^0$ | $10^1$ | $10^2$ | $10^3$ | $10^4$ |
| **True A** | **30** | **-0.30** | **-** | **-** | **0.070** | **0.038** | **0.179** | **0.300** | **-** |
| **A1** | 32.9±9.6 | -0.77±0.47 | - | - | 0.059 ±0.022 | 0.071 ±0.052 | 0.252 ±0.130 | 0.255 ±0.191 | - |
| **A2** | 32.0±9.8 | -0.72±0.45 | - | - | 0.062 ±0.021 | 0.067 ±0.053 | 0.286 ±0.132 | - | - |
| **A3** | 32.0±9.8 | -0.72±0.45 | - | 0.000 ±0.000 | 0.062 ±0.021 | 0.067 ±0.053 | 0.286 ±0.132 | - | - |
| **A4** | 34.0±9.2 | -0.70±0.46 | - | - | 0.062 ±0.021 | 0.082 ±0.050 | 0.191 ±0.084 | 0.259 ±0.198 | - |
| **True B** | **30** | **-0.30** | **0.001** | **0.012** | **0.037** | **0.088** | **0.099** | **0.250** | **0.800** |
| **B1** | 33.8±9.2 | -0.95±0.21 | - | - | 0.052 ±0.011 | 0.037 ±0.039 | 0.374 ±0.122 | 0.226 ±0.176 | - |
| **B2** | 36.5±7.6 | -0.93±0.26 | - | - | 0.050 ±0.000 | 0.051 ±0.039 | 0.292 ±0.103 | 0.234 ±0.196 | - |
| **True C** | **115** | **-2.02** | **0.118** | **0.094** | **0.116** | **0.247** | - | **-** | **-** |
| **C1** | 104.6±24.0 | -1.74±0.97 | 0.126 ±0.086 | 0.116 ±0.090 | 0.154 ±0.116 | 0.216 ±0.126 | - | - | **-** |
| **C2** | 91.2±19.2 | -2.36±0.83 | - | - | 0.415 ±0.099 | 0.143 ±0.117 | 0.137 ±0.113 | 0.115 ±0.095 | - |

**Table 2:** The *mean normilized error* (MNE) between the "measurements" and "true" values, and between the "measurements" and the model estimated values for the different tests.

| Test | "Measurements" vs "True" [a] | | | "Measurements" vs Estimated $MNE_M$ [b] | | |
|:---:|:---:|:---:|:---:|:---:|:---:|:---:|
| | **Yield** | **TD** | **Dilution** | **Yield** | **TD** | **Dilution** |
| **A1** | 21.2 | 7.6 | 9.4 | 25.0 | 7.0 | 16.69 |
| **A2** | 21.2 | 7.6 | 9.4 | 25.1 | 7.1 | 16.71 |
| **A3** | 21.2 | 7.6 | 9.4 | 25.1 | 7.1 | 16.71 |
| **A4** | 17.8 | 7.6 | 9.4 | 22.4 | 7.1 | 19.7 |
| **B1** | 20.5 | 6.9 | 5.6 | 20.6 | 6.0 | 14.7 |
| **B2** | 18.1 | 6.9 | 5.6 | 19.1 | 7.8 | 18.1 |
| **C1** | 8.4 | 11.6 | 1.8 | 6.3 | 12.9 | 3.5 |
| **C2** | 8.4 | 11.6 | 1.8 | 8.6 | 32.4 | 2.3 |

[a] Calculated by $\dfrac{100}{N_O}\sum_{i=1}^{N_O}\dfrac{\left|O_i - TR_i\right|}{O_i}$.

[b] Calculated by Eq. (16).

**Table 3:** The *mean normilized error* between the "true" and estimated values ($MNE_T$) for the different tests.

| Test | Yield | | | | TD | Dilution |
|------|-------|-------|-------|-------|------|----------|
|      | **5 ºC** | **15 ºC** | **25 ºC** | **35 ºC** |      |          |
| **A1** | 24.4 | 21.0 | 17.3 | 13.8 | 5.5 | 19.0 |
| **A2** | 21.4 | 19.5 | 16.9 | 14.1 | 4.7 | 18.5 |
| **A3** | 21.4 | 19.5 | 16.9 | 14.1 | 4.7 | 18.5 |
| **A4** | 20.4 | 18.3 | 15.7 | 12.9 | 6.0 | 22.5 |
| **B1** | 31.3 | 21.7 | 13.9 | 8.7 | 4.0 | 13.3 |
| **B2** | 24.4 | 15.6 | 9.0 | 6.4 | 2.5 | 18.4 |
| **C1** | 6.2 | 6.8 | 9.6 | 15.5 | 4.4 (110 ºC)[*] <br> 10.6 (140 ºC)[*] | 2.7 |
| **C2** | 18.1 | 9.6 | 7.2 | 11.5 | 9.0 (110 ºC)[*] <br> 27.8 (140 ºC)[*] | 3.4 |

[*] The errors for TD were calculated up to the denoted temperature in the parenthesis.

**Table 4:** The average relative standard deviation (*ARSD*) for the different tests.

| Test | Yield | | | | TD | Dilution |
|------|-------|-------|-------|-------|------|----------|
|      | **5 ºC** | **15 ºC** | **25 ºC** | **35 ºC** |      |          |
| **A1** | 34.6 | 29.7 | 26.0 | 24.2 | 21.0 | 23.6 |
| **A2** | 32.1 | 28.5 | 25.2 | 23.3 | 21.1 | 23.2 |
| **A3** | 32.1 | 28.5 | 25.2 | 23.3 | 21.1 | 23.2 |
| **A4** | 30.8 | 27.2 | 24.5 | 23.3 | 21.0 | 22.1 |
| **B1** | 37.1 | 27.2 | 20.0 | 16.9 | 20.5 | 18.0 |
| **B2** | 33.8 | 25.0 | 18.5 | 15.7 | 18.0 | 15.9 |
| **C1** | 15.0 | 14.9 | 16.2 | 22.9 | 20.7[*] | 16.5 |
| **C2** | 20.1 | 15.6 | 14.1 | 21.3 | 20.6[*] | 9.8 |

[*] The *ARSD* for the TD MFR values were calculated in the 20–120 ºC temperature range.

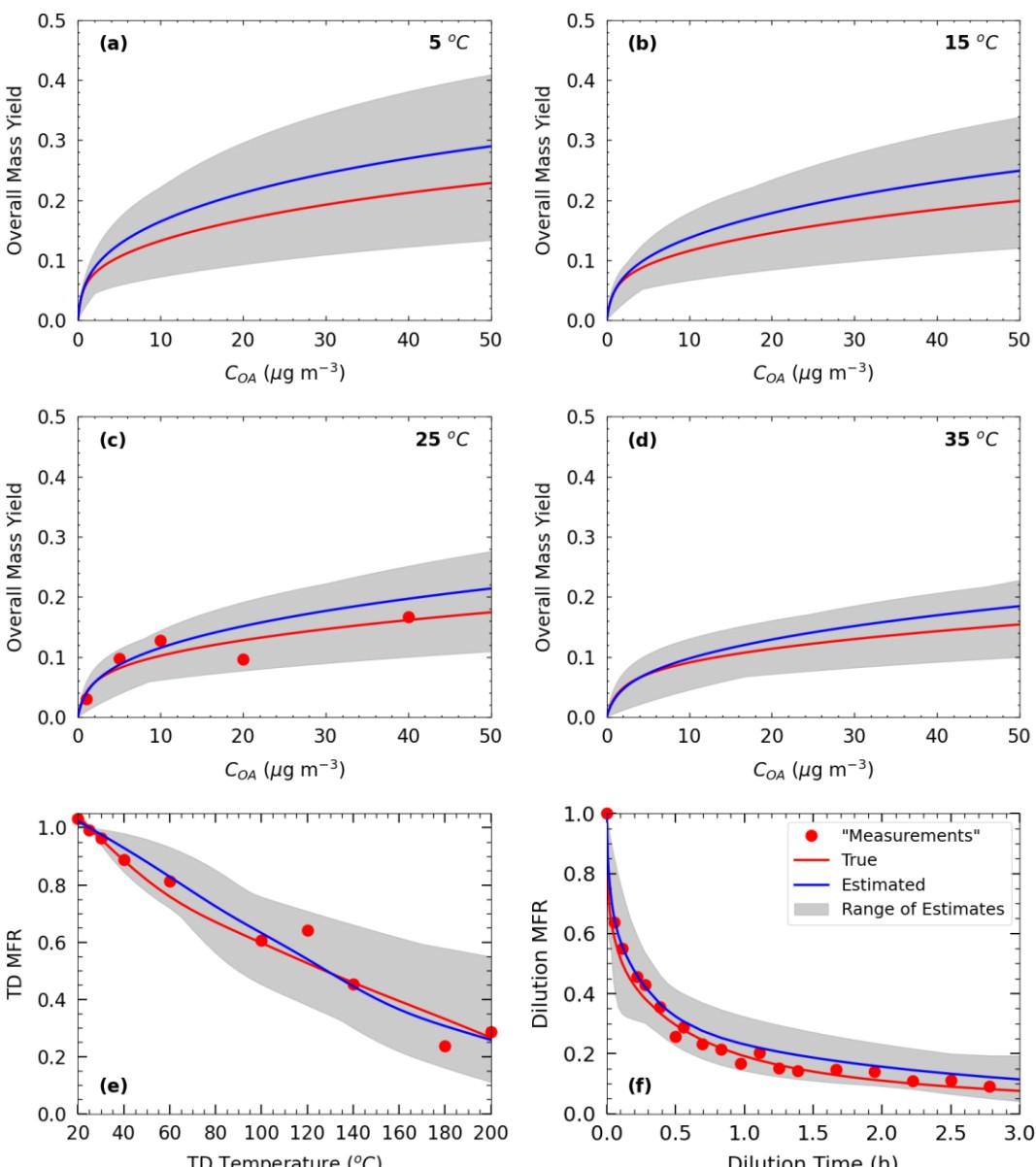

**Figure 1:** "Measurements" of Test A1 in Experiment A (red dots), true (red line) and estimated (blue line) yields at (**a**) 5 °C, (**b**) 15 °C, (**c**) 25 °C, and (**d**) 35 °C), (**e**) TD (thermogram), and (**f**) dilution (areogram) values. The grey area shows the range of good solutions obtained by our algorithm.

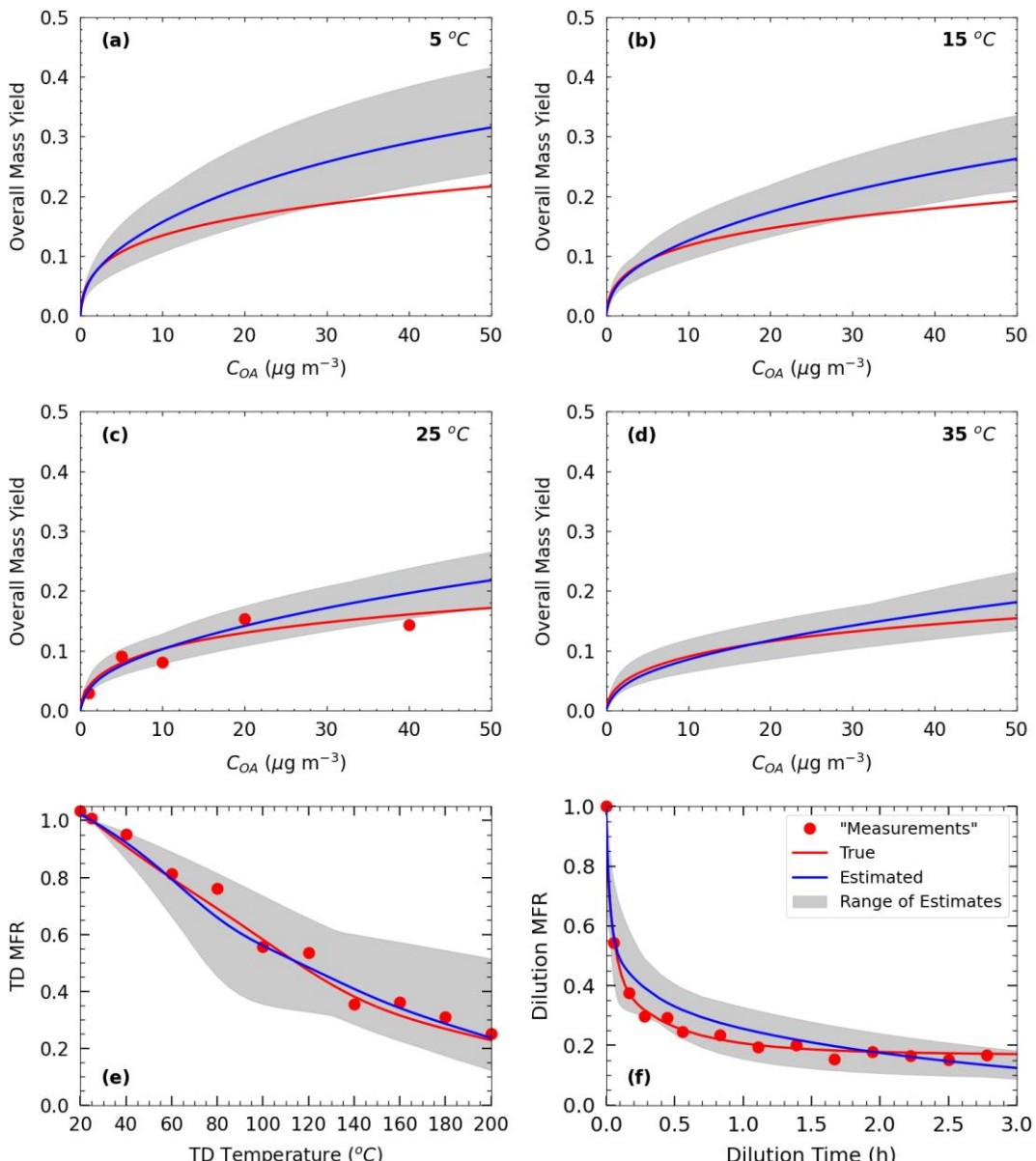

**Figure 2:** "Measurements" of Test B1 in Experiment B (red dots), true (red line) and estimated (blue line) yields at (**a**) 5 °C, (**b**) 15 °C, (**c**) 25 °C, and (**d**) 35 °C), (**e**) TD (thermogram), and (**f**) dilution (areogram) values. The grey area shows the range of good solutions.

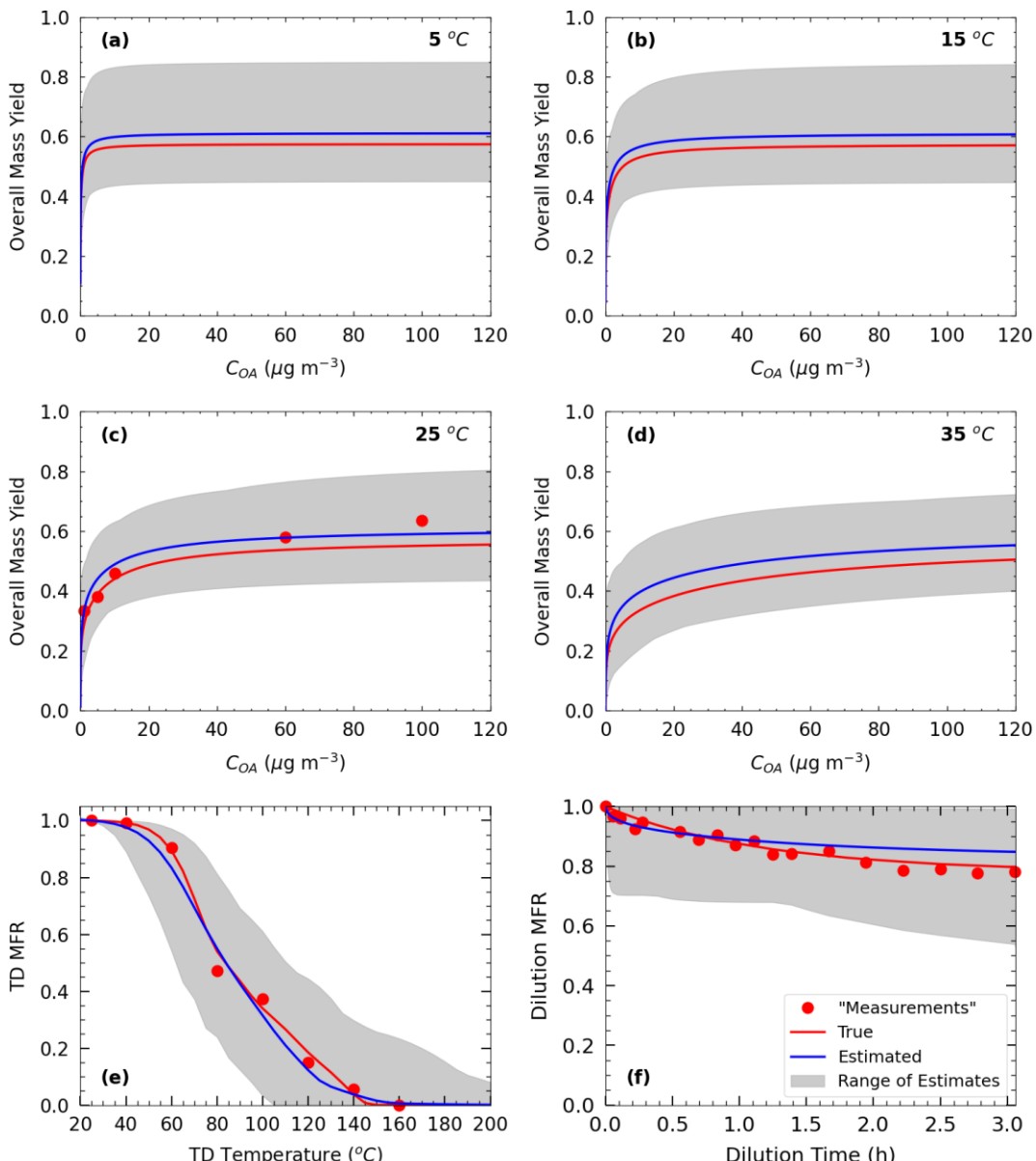

**Figure 3:** "Measurements" of Test C1 in Experiment C (red dots), true (red line) and estimated (blue line) yields at (**a**) 5 °C, (**b**) 15 °C, (**c**) 25 °C, and (**d**) 35 °C), (**e**) TD (thermogram), and (f) dilution (areogram) values. The grey area shows the range of good solutions.

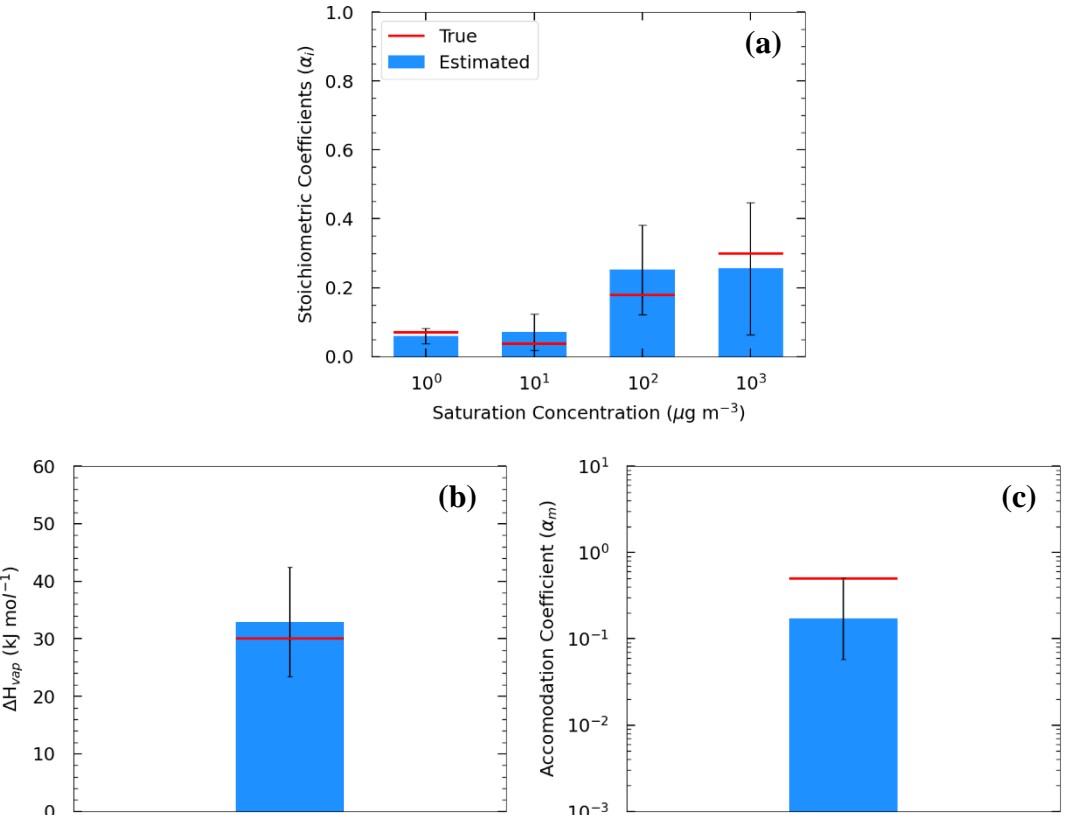

**Figure 4.** Estimated (bars) and true (red lines) parameter values of Experiment A in Test A1 combining yield, TD, and isothermal dilution measurements for: **(a)** the volatility distribution of the products, **(b)** $\Delta H_{vap}$, and **(c)** $\alpha_m$. The error bars represent the uncertainty of the estimated values.

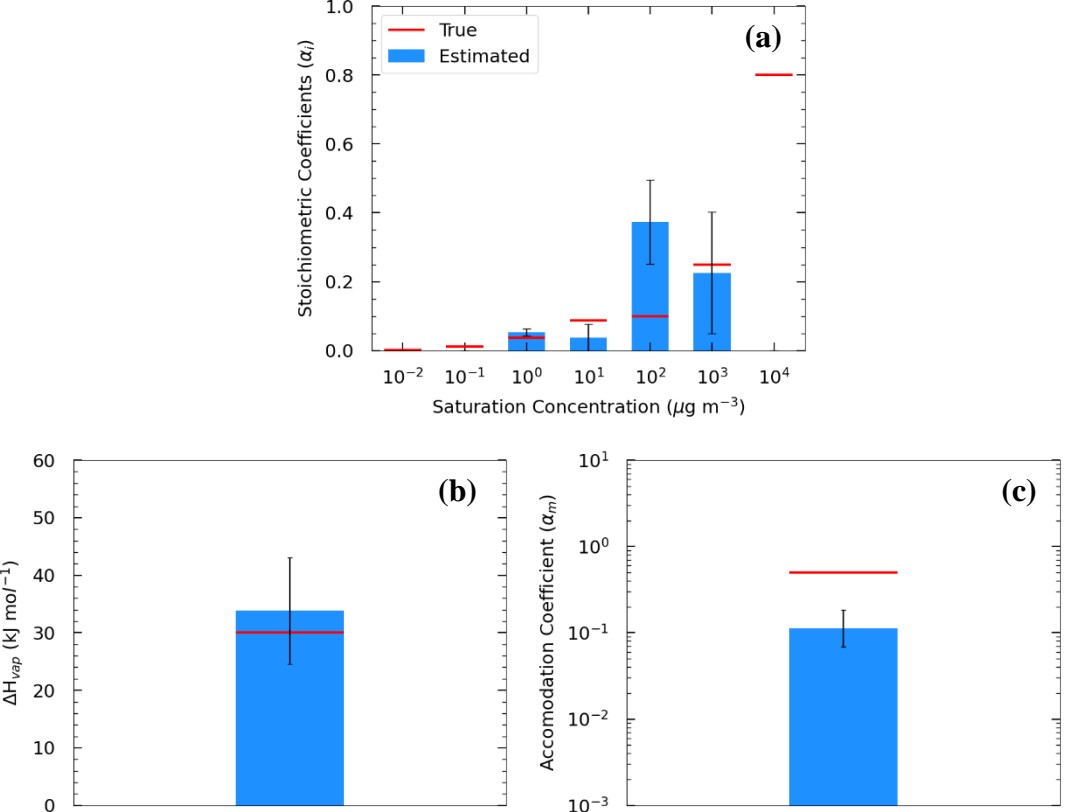

**Figure 5:** Estimated (bars) and true (red lines) parameter values of Experiment B in Test B1 combining yield, TD, and isothermal dilution measurements for: **(a)** the volatility distribution of the products, **(b)** $\Delta H_{vap}$, and **(c)** $\alpha_m$. The error bars represent the uncertainty of the estimated values.

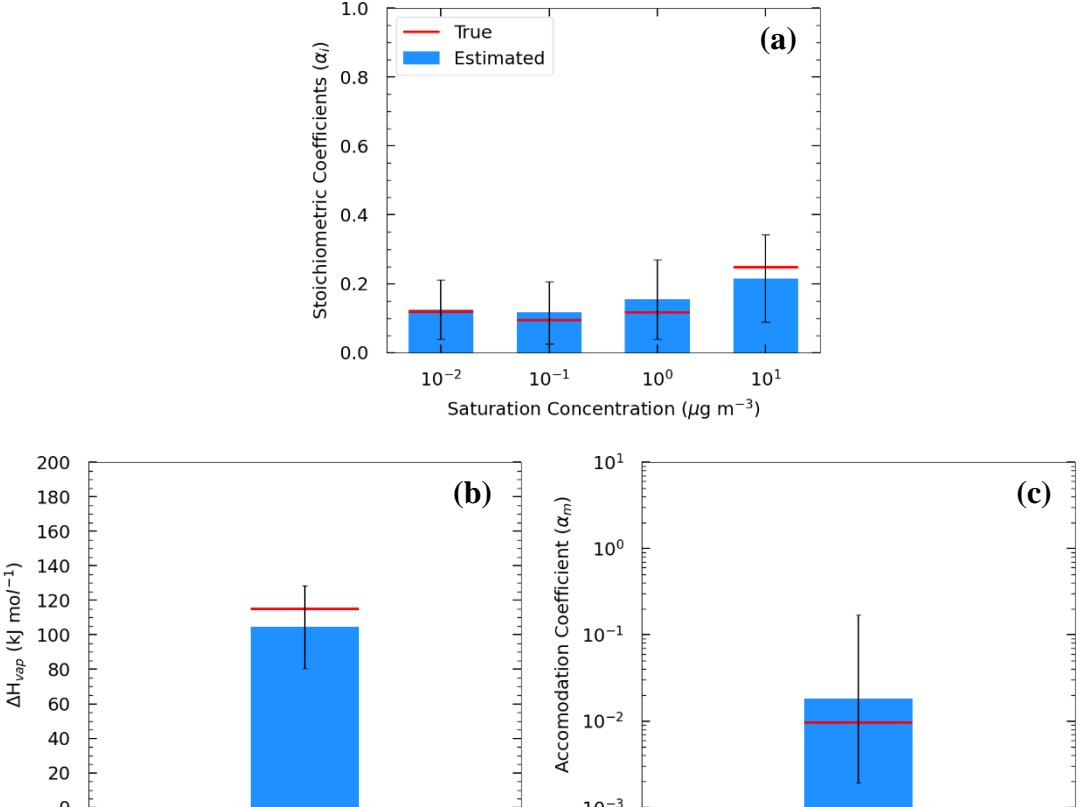

**Figure 6:** Estimated (bars) and true (red lines) parameter values of Experiment C in Test C1 combining yield, TD, and isothermal dilution measurements for: **(a)** the volatility distribution of the products, **(b)** $\Delta H_{vap}$, and **(c)** $\alpha_m$. The error bars represent the uncertainty of the estimated values.

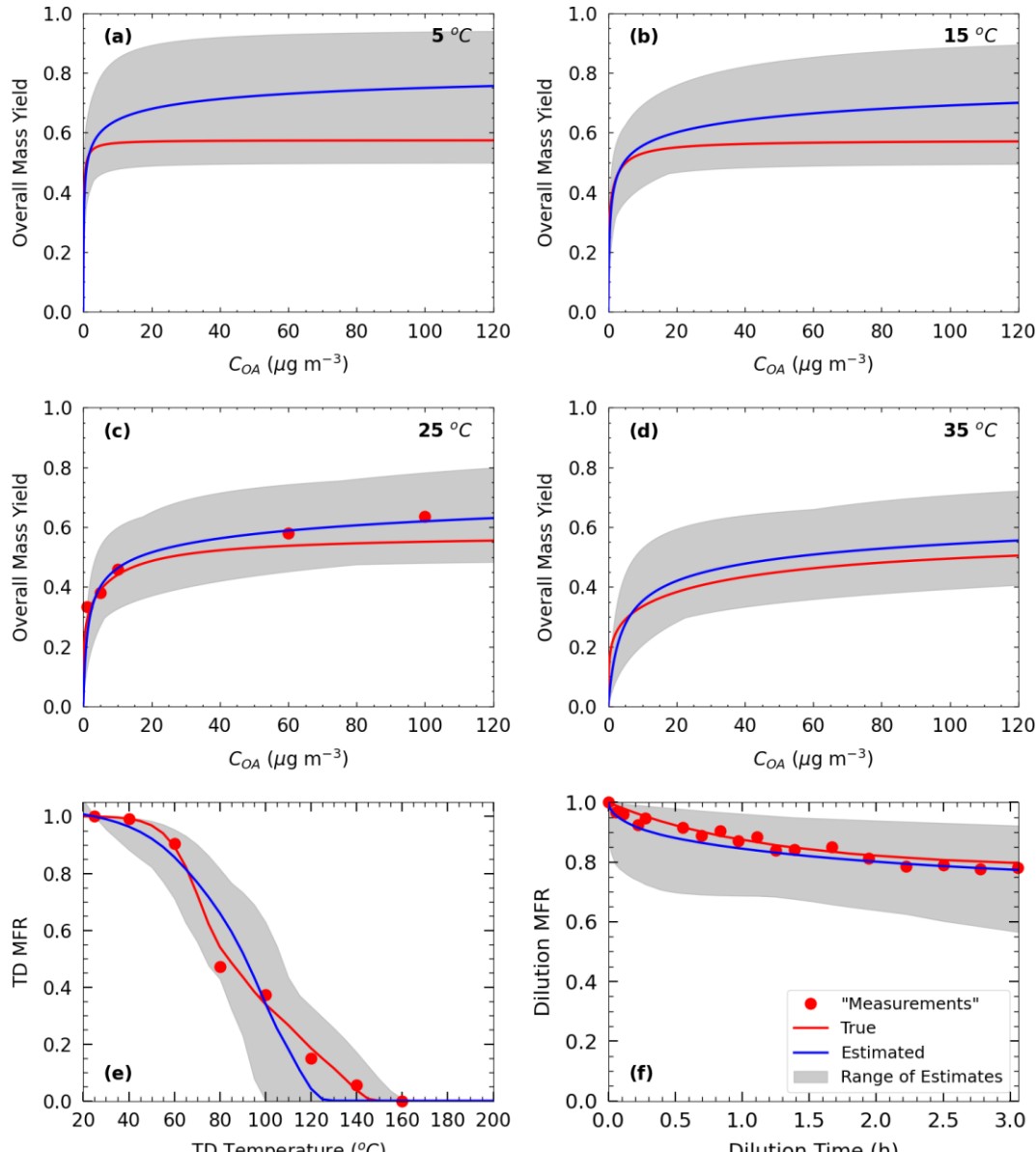

**Figure 7:** Yields calculated using the "true" parameters of Experiment C (red line) and the estimated (blue line) using the parameters of Test C2 for the following temperatures: (**a**) 5 °C, (**b**) 15 °C, (**c**) 25 °C, and (**d**) 35 °C. Also shown the (**e**) thermogram and (**f**) aerogram. The grey area shows the range of good solutions obtained by our algorithm.

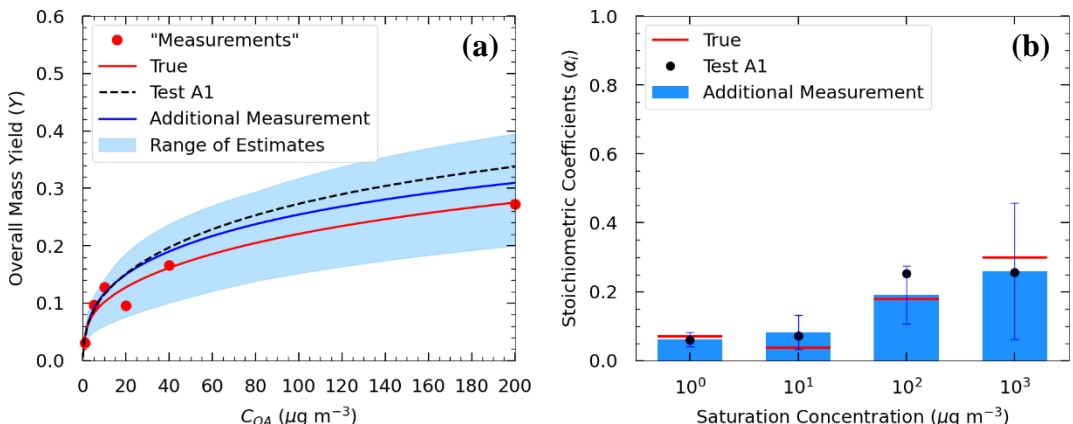

**Figure 8: (a)** True (red line) and estimated (blue line) yields in Test A4, and the "measurements" of Experiment A (red dots) including an additional yield "measurement" at 200 µg m$^{-3}$. The black dashed line corresponds to the estimated yields in Test A1. **(b)** Estimated volatility distribution of the products (bars) of Test A4 and the true (red lines) parameter values. The black dots correspond to the estimated volatility distribution of the products in Test A1.

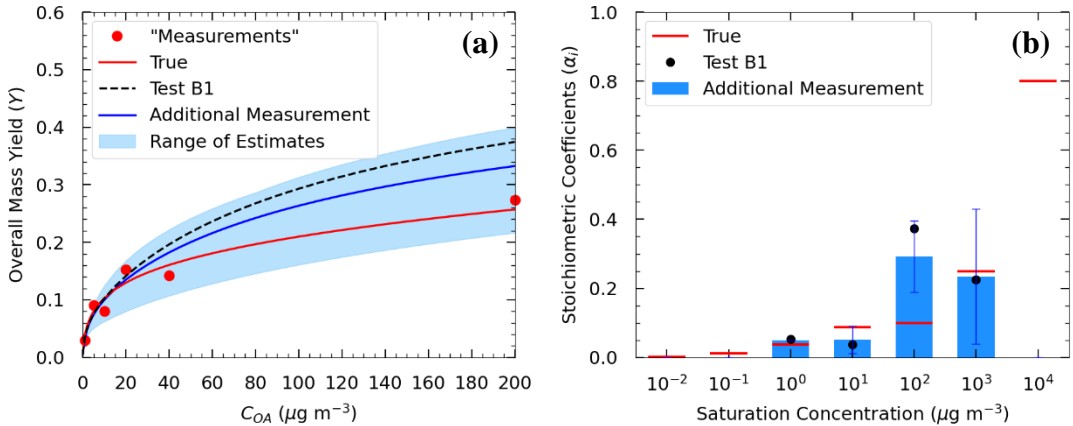

**Figure 9: (a)** Estimated yields (blue line) in Test B2 and "measurements" of Experiment B (red dots) including an additional yield "measurement" at 200 µg m$^{-3}$. The black dashed line corresponds to the estimated yields in Test B1. **(b)** Estimated volatility distribution of the products (bars) of Test B2 and the true (red lines) parameter values. The black dots correspond to the estimated volatility distribution of the products in Test B1.

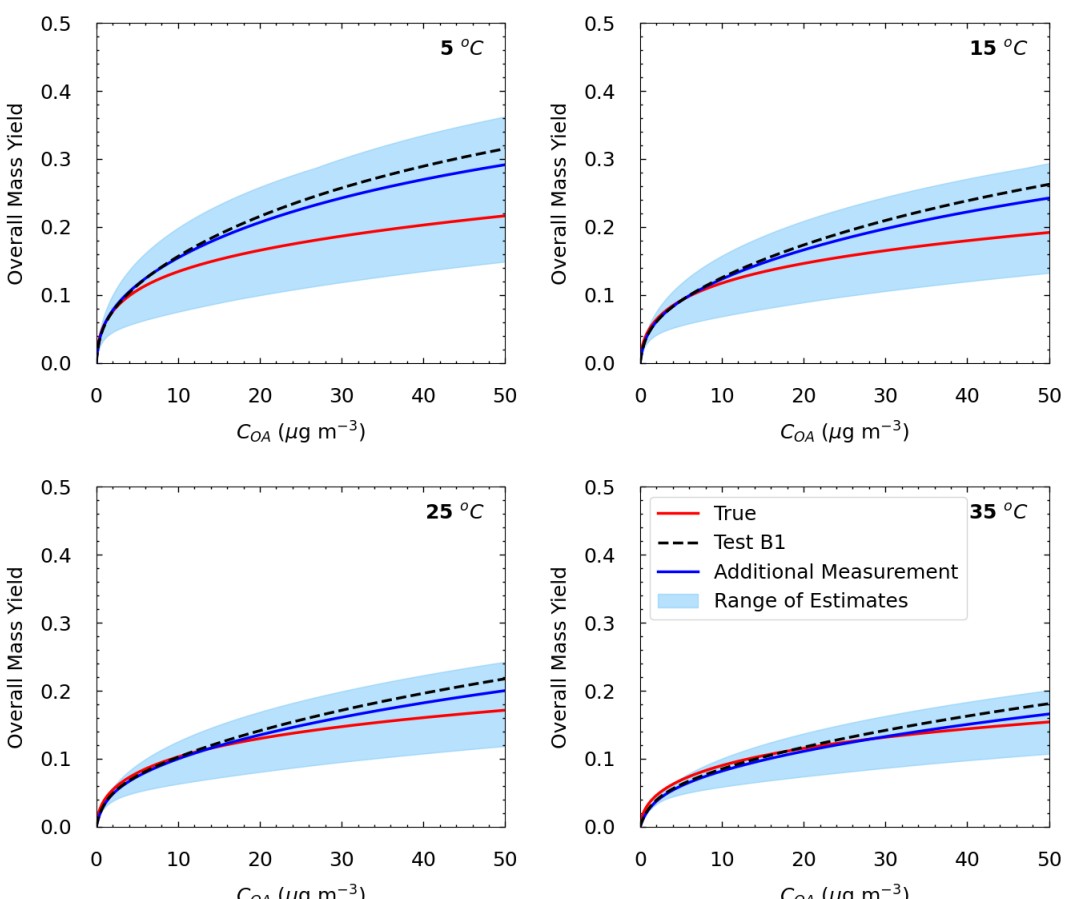

**Figure 10:** Yields calculated using the "true" parameters of Experiment B (red line) and the estimated (blue line) using the parameters of Test B2 for the following temperatures: 5 ℃, 15 ℃, 25 ℃, and 35 ℃. The blue area shows the range of good solutions obtained by our algorithm. The black dashed line corresponds to the estimated yields in Test B1.