# Peer review of "Estimation of secondary organic aerosol formation parameters for the Volatility Basis Set combining thermodenuder, isothermal dilution and yield measurements"

_Atmospheric Measurement Techniques, 2022_

## Referee Comment (RC2)

**Reviewer report Uruci et al., 2023, AMT**

Uruci et al. extended an existing algorithm to derive particle volatility information from the combination of thermal evaporation in a thermodenuder (TD), isothermal evaporation in a dilution chamber (DC), and yield experiments (YE). They used artificial data to evaluate the performance of their algorithm.

The topic is suitable for publication in AMT and highly relevant for a broad audience in atmospheric science. The overall presentation is good, and the descriptions are generally easy to follow. But there are two major issues that must be address prior to publication.

**Major comments**

1) From how I understand the generation of the artificial data set, the authors use circular reasoning when evaluating their algorithm.
In their algorithm, they compare the "measurement" data (yield curve, thermogram, and areogram) with a lookup table of yield curves, thermograms, and areograms generated for a large number of combinations of VBS, enthalpy of evaporation ($\Delta H_{evap}$,), and mass accommodation coefficient ($\alpha_M$) values. The calculated curve within 5% of the "measured" values are picked and the underlying VBS, $\Delta H_{evap}$, and $\alpha_M$ combinations are presented.
To generate "measurement" data for a known set of VBS, $\Delta H_{evap}$, and $\alpha_M$ values they start with values derived from yield experiments and then generate a thermogram and an areogram using the same thermodenuder and evaporation model as in their algorithm.
When they now compare these generated curves with the look up table, they will, of course, find good matches if the input parameters for the measured data (true VBS) were covered in the lookup table generation (see also specific comments 2 and 3). The good agreement and narrow range of the <5% solutions only shows that there is low ambiguity in the method. I.e., there are not many combinations of values far away from the true ones that produce matching yield, thermogram, and aerogram curves. In other words: if you use the values a, b, c to calculate thermograms and areograms, the algorithm will tell you that you used the values a, b, c if you included a, b, c when calculating your lookup table. To be blunt, the authors just showed that their equations work both ways. But they have not shown how well their method works with data that was not calculated with the model used in the algorithm.

2) The manuscript closes with the recommendation to use the combination of YE, TD, and DC data for future parametrisation. The manuscript did not convince me that the addition of YE data truly improves the results. Yes, the results using all three data sets look good. But nowhere do the authors show that their results are better than those from the method of Karnezi et al. (2014) with just TD and DC data. How much differ the results (combination of values for VBS distribution, $\Delta H_{evap}$, and $\alpha_M$) when only TD and DC data is combined vs using all three (TD, DC, and YE)? Unfortunately, the used data sets all derive the thermogram and areogram data from the yield data (see first major comment). Thus, I am not sure if this test will really be convincing or again simply show that the algorithm in itself is sound.
However, the authors need to present stronger arguments why the inclusion of YE data is beneficial, especially considering the much higher experimental effort needed to obtain YE data.

**Specific comments**

1) Line 189: Why was sum($\alpha_i$)<1 chosen as a criterium? There is no reason for that as $\alpha_i$ are stochiometric coefficients and not mass or mole fractions. The true VBS of case B shows a violation of that rule ($10^3$ + $10^4$ bin signal is already >1).

2) Following up on the previous comment: Because of this rule, the lookup table combinations do not cover the true VBS values in case B and more discrepancies are seen, especially in the areogram as that is most affected by the higher $C^*$ bins. It seems that the $\alpha_M$ value may be compensating the absence of the highest volatility bin somehow. This behaviour should be investigated as it has implications for the role of the $\alpha_M$ parameter in the algorithm which may not be desired.

3) Why was Case B only tested with 4 VBS bins? 60% of the signal is assigned to the $10^4$ bin which is not part of the lookup range. Comparing the estimated VBS distributions from case A and B one could argue that the estimations for Case B (blue, see Fig R1) are more similar to the true VBS distribution in case A (red) than to the true distribution of case B (yellow). Since the "true" VBS distributions in case A and B are derived from the same SOA data, one could come to the conclusion that the algorithm wants to find a solution close to the true case A VBS distribution. What are the authors thoughts on such reasoning? Would using more VBS bins (and including the $10^4$ bin) change this behaviour? I.e., would the algorithm suggest an estimate more similar to the "true" case B VBS values?

[Figure]

Figure R1: True and estimated VBS distributions for cases A1 and B1 visualised using values presented in Table 1 of the main manuscript.

4) Why was case A only tested for shifting to lower volatility bins? Especially, since the alternative parametrisation from the original paper (= case B) has very strong contributions from the $10^4$ bin. Also, what results would be obtained if the full range of VBS bins ($10^{-2}$ to $10^4$) was used?

5) Lines 482ff: The authors need to define more clearly what they mean by "robustness of their algorithm". Are they primarily interested in predicting yields? Then the algorithm indeed seems robust and reliable. But the tests with the shifted volatility range show that for case C a completely different VBS distribution recreates the yield curve and areogram as well as the true VBS distribution. This behaviour could be called being subjective to the choice of input parameters by the user – so very much not "robust". How would the user know if the yields are "right for the wrong reasons"? I.e., which volatility range would they choose without prior knowledge?
In the case C2, it seems that the lower $\Delta H_{evap}$ compensates the shift in volatility (again this should tell as something about the mechanism of the algorithm/model). It is hard to predict how these values will behave when they are used in a different context (e.g., new particle formation in regional model). They do distort the shape of the thermogram somewhat which could be used as an indicator for a "right for wrong reasons" case. But what would be objective criteria for "too much deviation" to identify a not that good solution?

6) After (hopefully) showing that the YE data really improves the predictions, I wonder if the combination of TD and YE data works as well as the combination of all three. I.e., do the DC and YE data sets essentially cover the same aspects of the underlying true values? The aim of the question is: What should be measured to obtain the most reliable VBS distribution, $\Delta H_{evap}$, and $\alpha_M$ combination? Especially when considering the time and effort needed for the measurements.

7) What about when the method is applied to real measurement data and that there was a process not covered in the model (e.g., the occurrence of thermal decomposition in the TD which shifts the thermogram towards apparent higher volatility). What would the algorithm make of that? Would the user be able to see that something is not right? Or would everything look fine, and the user will base their evaluation on incorrect VBS, $\Delta H_{evap}$, and $\alpha_M$ values?

8) Line 63ff and 325ff: The method by Stainer et al. and the assumptions for predicting the yield curves at different T ignores that with changing temperature the chemical formation pathways may change. Especially, HOM and/or dimer formation can be strongly affected and thus have a unexpected effect on the observed VBS and yield (e.g. Quelever et al., 2019; Gao et al., 2022). They authors should at least mention this aspect somewhere when discussing yield curves at different temperature.

9) Line 180ff: Should not the dilution ratio also play a role for the isothermal evaporation in a dilution chamber?

10) Line 190ff: The authors provide the number of combinations that need to be calculated. How does that translate to computational time/effort? Can this be run on an office PC at reasonable time? Can the lookup table be created once and then used for the comparison with multiple "measurements "?

11) Why was NMSE used to get the overall error to find the "closest" solutions, but in section 4.2 and later the solutions are compared using MNE?

12) Line 373ff: The wording here makes it sound as if the Experiment B data was from a completely different SOA system. But it is based on the same measured data as Experiment A. Only the number of VBS bins is changed. The authors should make this fact clearer.

13) Fig1 – 3 and S1 - S3: The information about the content of each panel in these figures is there. But labelling the panels with a, b, c etc in each figure will make it easier for the reader. Currently, only the Figure is referenced in the text and the reader has to figure out which of the panels is meant in the text.

14) When the authors compare the different cases, e.g., when adding the additional high $c_{OA}$ data point, it is difficult to judge how much the reconstructed VBS distribution, yield curves, etc. really change from the base case. E.g., Fig 8a needs to be compared with the 25 °C panel in Fig 1 which has different x and y axis scaling. It would be very helpful to add the base case lines/bars to Fig 8 and 9. Especially the extrapolation of the base case to 200 ug m$^{-3}$ should provide an even stronger argument why the extra data point is useful.

15) I need more information about the weighted averaging of the selected <5% solutions. When the averages are calculated, is each data point treated individually? I.e., data point 1 in solution 1 is close to the true data and gets a high weight. But data point 3 of solution 1 is far away (i.e., the slope of the solution is wrong). Does data point 3 then get a different weight than data point 1? Or do all data points of a solution get the same weight factor?

16) How many solutions usually are in the <5% group? Did this number differ between the investigated cases? E.g., were there less acceptable solutions when the "wrong" VBS range was chosen? If data points were treated individually, how much did the number of <5% solutions vary for the data points?

17) Following up on the previous comment assuming that each data point is treated individually: The authors could consider improving the visualisation of the range of estimates. Instead of a uniform grey area, they could indicate the density of solution curves with a colour scale. That would preserve the range of slopes of the solutions. Two examples of such figures are shown below.

[Figure]

Figure R1: Lower half of Fig 2 in Li et al. (2019). Enthalpy of evaporation ($\Delta H_{vap}$) values from the best-fit simulations are presented with circles and the parameter density with shades of grey. The parameter density is derived by dividing $\Delta H_{vap}$ and log $C^*$ space into grid cells, counting the frequency of simulated parameters inside each cell, and normalizing to the sum of frequency in each log $C^*$ column.

[Figure]

**Figure S1**: Relative evaporation curve density of the three different optimization schemes applied to artificial data set 1. The relative evaporation curve density is calculated by dividing the EF and time space to grids and counting how many of the simulated evaporation curve goes through a particular grid box. The counts are then normalized by the highest count in every time column. White color indicates that no simulation goes through that area. a) MCGA method with uniform sampling of the parameter space. b) MCGA with sampling similar to the Bayesian inversion method (see main text, Sect. 3) c) Bayesian inference method.

Figure R2: Figure S1 from Tikkanen et al., (2019)

**Typos and language comments**

1) Line 29: "The predicted yield uncertainty…" I find this sentence hard to understand.
2) Line 35: "IPCC, 2013" should be updated to IPCC, 2021 unless the authors are referring to something very specific which is only contained in the 5th assessment report.
3) Line 103: comma before "respectively"
4) Line 184: "SOA partitioning model" does that refer to the Eq. 1-3 in section 2.1?
5) Eq. 9: One sum goes up to "$N_0$" the other to "n". Are these indeed different numbers or is it a typo?
6) Line 219: make it clear that these are the same models that are used by the analysis algorithm.
7) line 365: "wider range" is not a precise description here. Better say "at higher SOA concentrations". Wider could also mean extending the range to lower concentrations.
8) Line 304 "(20 to 200 °C with a step of 5 °C but including TD MFR values greater than zero)" I do not understand the "but…" part. If "but only including" was meant, how could there even be MFR values <0?
9) Line 306 "[…] we used a higher resolution for the first 0.5h (step of 2 min), in which the evaporation is usually faster, and lower then (step of 10 min) up to 3 hours." What is meant with the "lower then …" part?

Line 445 "[…] instead of the 4 bins used in Test A1 and covering the $10^3$ µg m$^{-3}$ material." I am not sure about the meaning of the underlined part in this sentence.

---

## Referee Comment (RC3)

Uruci et al. extended an algorithm by combining SOA yield measurements, thermodenuders (TD), and isothermal dilution (TD) to constrain the volatility product distribution ($\alpha_i$), effective vaporization enthalpy ($\Delta H_{vap}$), and effective accommodation coefficient ($\alpha_m$) and finally reduce the uncertainty of parameterization of SOA formation.

The topic is worthy of investigation as the parameterization of SOA is a significant source of uncertainty in air quality modeling and is also suitable for publication in AMT. The overall writing is clear. However, there are some weaknesses in the result and discussion part that needed to be concerned prior to publication.

**Major comments:**

1. The main purpose of the study is to combine SOA yield measurements with TD and ID to reduce the uncertainties of estimation of $\alpha_i$, $\Delta H_{vap}$, and $\alpha_m$. However, there is no comparison of estimated parameters between derived from combining approach tests and derived from single-set tests to indicate the improvement.

2. The estimates of SOA yields, TD, and ID in all cases were good with $MNE_M$ of ~25% or less (Table 2), but the estimated $\alpha_m$ dramatically deviated from the truth value for almost all tests (Table 1). It indicates 'the difficulty of constraining $\alpha_m$.' Does this mean that the extended approach fails to improve the estimation of parameters, or at least for accommodation coefficients? In other words, why should we focus on the accommodation coefficients to decrease the uncertainty of parameterization of SOA yields?

3. For the description of Experiment B1 in section 4.2.2, the saturation concentrations of LVOCs, SVOCs, and IVOCs range from $10^{-2}$ to $10^4$ μg m$^{-3}$, but the SOA yields of α-pinene (precursor for Experimental B) are quite different from them. I wonder whether it is applicable to use a larger volatility bin for this oxidation system, as Pathak et al. (2007b) reported that 'the performance of the both 4- and 7-product basis set parameterizations is similar for practically all data points.'

4. Table 2 lists mean normalized errors of measurement data vs 'True' values ($MNE_T$) and vs estimated values ($MNE_M$). I wonder if the authors try to use similar values of $MNE_T$ and $MNE_M$ to support the good performance of the extended approach. And what is the reason for causing the small difference between them?

5. What do 'close to unity' and 'resistances to mass transfer are small' mean in Line 370? The authors may want to include a description of the accommodation coefficient in the introduction section.

**Language suggestion:**

1) Line 20: add a comma between 'approach' and 'we';

2) Line 29: 'less' should be changed to 'smaller';

3) Line 74: give the whole word when first using the abbreviation for 'TD', and also for 'LVOCs' in Line 112;

4) Line 77: delete 'more';

5) Line 85-88: The sentence 'In TD applications in the …' is hard to understand, please rewrite it;

6) Line 100: delete 'two';

7) Line 101: 'type' should be changed to 'types';

8) Line 106: delete 'of';

9) Line 107: 'obtained' should be changed to 'conducted';

10) Line 114: add a comma between 'enthalpy' and 'we';

11) Line 119: replace 'so' with 'thus' or 'therefore', and change throughout the text;

12) Line 148: change the sentence 'The time-dependent evaporation of SOA in the TD is described in this work…' to 'The time-dependent evaporation of SOA in the TD used in this work is described…'

13) Line 288: delete 'also against';

14) Line 403: add a comma between 'problems' and 'the';

15) Line 457: Please simplify the sentence '…covering in that way…' to make it easy to understand.

---

## Author Response (AR1)

**Responses to the Comments of Reviewer 1**

**(1)** The authors developed an algorithm to estimate the SOA product volatility distribution, effective vaporization enthalpy, and effective accommodation coefficient combining SOA yield measurements from different experiments. Also, the approach can also estimate the uncertainty of the predicted yields for different atmospheric conditions. Overall, the approach is useful and meaningful for the deeper analysis of SOA formation. The evaluation of the approach presented in the current manuscript is comprehensive and convincing.

We appreciate the positive assessment of our work by the reviewer. Our responses (in black) follow each comment of the reviewer (in blue).

**(2)** In the section 4 of Testing of the Algorithm, detailed simulation results were provided. Figures showed generally good agreement between true /measurement values and estimated values, and the discrepancies seem mostly below 30%. Is it possible to add some references as a comparison? For example, what are the differences between true and estimated values in other methods/experiments in previous similar studies? Is <30% good enough or just average performance for model simulation on this purpose or pathway? Please give more literature references in section 4, which may help audiences better understand the significance of the work.

We have extended our discussion of the results of previous efforts to estimate SOA formation parameters from measurement data. Only a few of these studies used an approach similar to ours that is generating data from an SOA formation model and then introducing experimental error and as a result could quantify their error. The performance of the current algorithm appears to be better than in previous efforts mainly because it uses more measurements from different experimental techniques.

**(3)** I would recommend this paper be accepted after minor revision.

We appreciate it.

**Responses to the Comments of Reviewer 2**

**(1)** Uruci et al. extended an existing algorithm to derive particle volatility information from the combination of thermal evaporation in a thermodenuder (TD), isothermal evaporation in a dilution chamber (DC), and yield experiments (YE). They used artificial data to evaluate the performance of their algorithm. The topic is suitable for publication in AMT and highly relevant for a broad audience in atmospheric science. The overall presentation is good, and the descriptions are generally easy to follow. But there are two major issues that must be address prior to publication.

We would like to thank the reviewer for the constructive comments, which are very helpful for improving the clarity of the manuscript. Our responses (in black) follow each comment of the reviewer (in blue).

*Major comments*

**(2)** From how I understand the generation of the artificial data set, the authors use circular reasoning when evaluating their algorithm. In their algorithm, they compare the "measurement" data (yield curve, thermogram, and areogram) with a lookup table of yield curves, thermograms, and areograms generated for a large number of combinations of VBS, enthalpy of evaporation ($\Delta H_{evap}$) and mass accommodation coefficient ($\alpha_m$) values. The calculated curve within 5% of the "measured" values are picked and the underlying VBS, $\Delta H_{evap}$, and $\alpha_m$ combinations are presented. To generate "measurement" data for a known set of VBS, $\Delta H_{vap}$, and $\alpha_m$ values they start with values derived from yield experiments and then generate a thermogram and an areogram using the same thermodenuder and evaporation model as in their algorithm. When they now compare these generated curves with the look up table, they will, of course, find good matches if the input parameters for the measured data (true VBS) were covered in the lookup table generation (see also specific comments 2 and 3). The good agreement and narrow range of the <5% solutions only shows that there is low ambiguity in the method. I.e., there are not many combinations of values far away from the true ones that produce matching yield, thermogram, and aerogram curves. In other words: if you use the values a, b, c to calculate thermograms and areograms, the algorithm will tell you that you used the values a, b, c if you included a, b, c when calculating your lookup table. To be blunt, the authors just showed that their equations

work both ways. But they have not shown how well their method works with data that was not calculated with the model used in the algorithm.

This is an important aspect of our algorithm testing that was not made clear enough in our presentation. The results of the model are "corrupted" with experimental error before they are used for the fitting. This for example can be seen in Figure 1 in which the "measurements" can be quite different from the model predictions ("true") depending on the type of measurement. This means that there is no set of model parameters that can reproduce all the "measurements". This error introduction does break to a large extent the circular reasoning mentioned by the reviewer. This issue is now discussed in detail in the revised paper.

To address the second point (about use of data not calculated by the same model) we can only use the algorithm in realistic experimental data. We have done this in Sippial et al. (2023) using β-caryophyllene SOA measurements (yields, evaporation in a thermodenuder and isothermal dilution). However, the only measure of success of the algorithm in this case (when the actual processes and the true parameter values are unknown) is the discrepancy between model predictions and measurements. A brief discussion of this application has also been added to the paper.

**(3)** The manuscript closes with the recommendation to use the combination of YE, TD, and DC data for future parameterization. The manuscript did not convince me that the addition of YE data truly improves the results. Yes, the results using all three data sets look good. But nowhere do the authors show that their results are better than those from the method of Karnezi et al. (2014) with just TD and DC data.

How much differ the results (combination of values for VBS distribution, $\Delta H_{vap}$, and $\alpha_m$) when only TD and DC data is combined vs using all three (TD, DC, and YE)? Unfortunately, the used data sets all derive the thermogram and areogram data from the yield data (see first major comment). Thus, I am not sure if this test will really be convincing or again simply show that the algorithm in itself is sound.

However, the authors need to present stronger arguments why the inclusion of YE data is beneficial, especially considering the much higher experimental effort needed to obtain YE data.

This is a valid concern. We have followed the advice of the reviewer and rerun the algorithm for tests A1, B1, and C1 without providing as input the corresponding SOA yield measurements. In this way the algorithm is practically the same as that of Karnezi et al. (2014). In all cases, the absence of the yield measurements led to a significant deterioration of the ability of the algorithm to estimate SOA yields at all temperatures and concentrations. For example, in test A1 the SOA yield error of the algorithm in the 5-35 $^{\circ}$C temperature range increased from 14-24% to approximately 100%. The corresponding uncertainty range also increased by a factor of 4-6. Similar results were obtained in the other tests.

We have added a new section in the revised paper where we compare the results of the new algorithm to those of the original and demonstrate the significant improvement in the ability to estimate SOA yields at different conditions.

**Specific comments**

**(4)** Line 189: Why was sum($\alpha_i$)<1 chosen as a criterium? There is no reason for that as $\alpha_i$ are stoichiometric coefficients and not mass or mole fractions. The true VBS of case B shows a violation of that rule ($10^3 + 10^4$ bin signal is already >1).

We clarify in the revised paper that this upper limit is a parameter that can be selected by the user of the algorithm. While a sum of unity is a reasonable choice for oxidation products in the volatility bins less than 1000 μg m$^{-3}$, the sum can exceed this value. To test the sensitivity of our results we have repeated certain tests using 2 as an upper limit. The changes in the results of all tests were minor. We have added the corresponding discussion of the results of this sensitivity test of the algorithm in the revised paper.

**(5)** Following up on the previous comment: Because of this rule, the lookup table combinations do not cover the true VBS values in case B and more discrepancies are seen, especially in the areogram as that is most affected by the higher $C^*$ bins. It seems that the $\alpha_m$ value may be compensating the absence of the highest volatility bin somehow. This behaviour should be investigated as it has implications for the role of the $\alpha_m$ parameter in the algorithm which may not be desired.

The exploration of these interactions was indeed the reason for the design of test B. In this test we attempt to model the behavior of the system with a narrower volatility range than the real one. Of course, we are expecting errors in the estimation of the corresponding parameters due to this mismatch. The underestimation of the

accommodation coefficient is one of these errors. We agree with the reviewer that the model compensates for the missing volatility bins by increasing the material in the $10^2$ µg m$^{-3}$ bin and by decreasing the accommodation coefficient. However, we think that the most important result here is that even in this case the predicted yields at all conditions are within 20% of the true values. We have added this discussion in the paper.

**(6)** Why was Case B only tested with 4 VBS bins? 60% of the signal is assigned to the $10^4$ bin which is not part of the lookup range. Comparing the estimated VBS distributions from case A and B one could argue that the estimations for Case B (blue, see Fig R1) are more similar to the true VBS distribution in case A (red) than to the true distribution of case B (yellow). Since the "true" VBS distributions in case A and B are derived from the same SOA data, one could come to the conclusion that the algorithm wants to find a solution close to the true case A VBS distribution. What are the authors thoughts on such reasoning? Would using more VBS bins (and including the $10^4$ bin) change this behaviour? I.e., would the algorithm suggest an estimate more similar to the "true" case B VBS values?

This is an interesting point. The smog chamber data in Case B cover organic aerosol concentrations up to 40 µg m$^{-3}$. Even for this experiment, organic compounds with C*=$10^4$ µg m$^{-3}$ will be almost exclusively in the gas phase, and they will not contribute to the SOA concentration. Therefore, it would be for all practical purposes impossible to derive any useful information about these IVOCs from this experimental data set. Experiments at much higher concentrations (a few hundred µg m$^{-3}$) would be needed to start constraining these oxidation products based on the specific types of measurements. The objective of this test was indeed to quantify the effects of the mismatches between the true and retrieved volatility distribution. A brief discussion of the above issues has been added to the paper.

By comparing Tests B1 and B2 with the Case True A, one would indeed expect that the retrieved volatility distribution of the products will be similar because the pseudo data were extracted by using the parametrizations derived from the same smog chamber experiments. This is now mentioned in the paper.

**(7)** Why was case A only tested for shifting to lower volatility bins? Especially, since the alternative parametrization from the original paper (= case B) has very strong contributions from the $10^4$ bin. Also, what results would be obtained if the full range of VBS bins ($10^{-2}$ to $10^4$) was used?

The smog chamber data set in Case A also cover organic aerosol concentrations up to 40 μg m$^{-3}$. Therefore, the same argument presented in our response to Comment 6 applies here too. Given that all experiments took place in moderate SOA concentration levels it is practically impossible to constrain the $10^4$ bin with this information. The use of such a bin in this case provides little useful information and increases the uncertainty of all estimates. The use of a 7-volatility bin fit is expected to provide a better solution because it matches the true parameters. However, this is something that we have tried to avoid in these tests to minimize the extent of the circular reasoning problem mentioned by the reviewer in Comment 2. We should also point out that obtaining 7-bin solutions is quite computationally consuming and the resulting volatility distributions are quite uncertain because too many parameters are used to fit three relatively simple (monotonically varying) data sets. A few sentences have been added to the paper to discuss these issues.

**(8)** Lines 482ff: The authors need to define more clearly what they mean by "robustness of their algorithm". Are they primarily interested in predicting yields? Then the algorithm indeed seems robust and reliable. But the tests with the shifted volatility range show that for case C a completely different VBS distribution recreates the yield curve and areogram as well as the true VBS distribution. This behaviour could be called being subjective to the choice of input parameters by the user – so very much not "robust". How would the user know if the yields are "right for the wrong reasons"? I.e., which volatility range would they choose without prior knowledge?

In the case C2, it seems that the lower $\Delta H_{\text{evap}}$ compensates the shift in volatility (again this should tell as something about the mechanism of the algorithm/model). It is hard to predict how these values will behave when they are used in a different context (e.g., new particle formation in regional model). They do distort the shape of the thermogram somewhat which could be used as an indicator for a "right for wrong reasons" case. But what would be objective criteria for "too much deviation" to identify a not that good solution?

We agree that this issue needs clarification. Indeed, our statement refers to the prediction of yields in chemical transport models which is the single most important use of the derived VBS parameterizations. We show in this work that the estimated volatility distributions are more uncertain than the yield estimates. We also show that the estimates of the enthalpy of vaporization are also robust, while the estimates of the accommodation coefficients are very uncertain. The use of the results of these experiments that have been designed for the measurement of SOA yields to other applications (e.g., new particle formation) should be done with caution. Different experiments should be probably performed for the derivation of the VBS distribution in this case focusing on low concentration levels and the least volatile SOA components. We have added the corresponding discussion to the paper.

**(9)** After (hopefully) showing that the YE data really improves the predictions, I wonder if the combination of TD and YE data works as well as the combination of all three. I.e., do the DC and YE data sets essentially cover the same aspects of the underlying true values? The aim of the question is: What should be measured to obtain the most reliable VBS distribution, $\Delta H_{evap}$, and $\alpha_m$ combination? Especially when considering the time and effort needed for the measurements.

We have followed the suggestion of the reviewer and after showing that addition of the yield measurements improves the model predictions (see our response to Comment 3 above) we repeated selected tests using all combinations of the measurements (Yields/TD, Yields/Dilution and TD/Dilution). The Yield-Thermodenuder combination gave the best results out of the three pairs. The isothermal dilution measurements are the least valuable because only a relatively small fraction of the SOA evaporates and therefore the information provided is relatively limited and focuses on the more volatile components of the particles. Also, TD measurements are important to constrain well $\Delta H_{vap}$ and allow the more accurate extrapolation of the results to other temperatures. also provides information for the volatility distribution of the OA. The results of our tests (YE+TD, YE+DIL, TD+DIL) for Case A1 have been added to the Supplementary Information and their discussion to the main paper.

**(10)** What about when the method is applied to real measurement data and that there was a process not covered in the model (e.g., the occurrence of thermal decomposition in the TD which shifts the thermogram towards apparent higher volatility). What would

the algorithm make of that? Would the user be able to see that something is not right? Or would everything look fine, and the user will base their evaluation on incorrect VBS, $\Delta H_{evap}$, and $\alpha_m$ values?

One expects that when the model used to simulate a series of processes (SOA production, evaporation during heating, and isothermal evaporation) has a serious weakness (e.g., misses a process dominating the results) that the model would not be able to reproduce all observations. It is though possible that the missing process would not create a major change in the behavior of the system (e.g., an abrupt change in the slope of the thermogram) and that the model would be able to fit the results accounting indirectly for it. Thermal decomposition in the TD is such a process and could lead to overestimation of the volatility of the least volatile components of the SOA. This can make the quantification of LVOCs and ELVOCs quite uncertain with the techniques discussed here. On the other hand, the corresponding parameters for the more volatile LVOCs and the SVOCs that are important for atmospheric SOA modeling should be a lot less uncertain given that they are measured in relatively low TD temperatures. A brief discussion of this point has been added to the paper.

**(11)** Line 63ff and 325ff: The method by Stainer et al. and the assumptions for predicting the yield curves at different $T$ ignores that with changing temperature the chemical formation pathways may change. Especially, HOM and/or dimer formation can be strongly affected and thus have an unexpected effect on the observed VBS and yield (e.g., Quelever et al., 2019; Gao et al., 2022). They authors should at least mention this aspect somewhere when discussing yield curves at different temperature.

We do agree with the reviewer that the SOA formation is not only thermodynamically driven, but also kinetically. The current VBS parameterizations assume that the stoichiometric coefficients ($\alpha_i$) are temperature independent. While the corresponding dependency may be small for SVOCs, this may not be the case of components of low volatility like HOMs and dimers. This point and the corresponding references have been added to the paper.

**(12)** Line 180ff: Should not the dilution ratio also play a role for the isothermal evaporation in a dilution chamber?

Of course, the dilution ratio is an important parameter for the corresponding experiments. Unfortunately, the range of dilution ratios that can be used is rather limited

(usually 10-20). Low dilution ratios result in little evaporation and little signal to be explored. High dilution ratios lead to very low initial concentrations and a lot of noise in the subsequent measurements. A dilution ratio of 10 was assumed for Experiments A and B (Table S2), while for Experiment C we used the experimental value (dilution ratio of 17) based on the work of Sippial et al. (2023). When the SOA samples are injected in the chamber with a volume of clean air, the initial gas and aerosol concentrations are lowered by this factor, shifting the system out of equilibrium. This is now explained in the paper.

**(13)** Line 190ff: The authors provide the number of combinations that need to be calculated. How does that translate to computational time/effort? Can this be run on an office PC at reasonable time? Can the lookup table be created once and then used for the comparison with multiple "measurements "?

The computational cost depends mainly on the discretization of the stoichiometric mass yields ($\alpha_i$), $\Delta H_{vap}$, and $\alpha_m$ and the resolution in the TD temperatures (5 ºC in all tests). For a 4-product system resulting in 3,153 combinations of $a_i$ the CPU time was approximately 15 h in an office PC. So, it can be run in reasonable time, indeed a lot less than the time required to analyze the corresponding experimental data. This is now mentioned in the paper.

Evaporation in TD depends on the initial SOA mass, the mean volume diameter, and the residence time in the heating tube. Because these three quantities vary between experiments and are used as inputs to the algorithm, one needs to repeat the simulation for the specific experiment. Similarly, evaporation in the dilution chamber depends on the initial SOA mass, the mean volume diameter, and the dilution ratio, which again are inputs to the model to extract the corresponding areogram for every combination. The simulation of the TD and isothermal dilution measurements are the most CPU demanding processes in this algorithm. So given the number of degrees of freedom in the system, unfortunately it is not feasible to create the corresponding look-up tables.

**(14)** Why was NMSE used to get the overall error to find the "closest" solutions, but in section 4.2 and later the solutions are compared using MNE?

We have chosen the *NMSE* as the loss function of the algorithm, following standard practice to choose a differentiable function. This allows use of a series of minimization algorithms, which unfortunately are of little use in this specific problem due to the

multiple minima that are present. We have chosen the MNE for the presentation of the error of the various solution because it has a simple physical meaning and conveys easily to the readers how good (or bad) a given solution is. This explanation has been added to the paper.

**(15)** Line 373ff: The wording here makes it sound as if the Experiment B data was from a completely different SOA system. But it is based on the same measured data as Experiment A. Only the number of VBS bins is changed. The authors should make this fact clearer.

We have followed the advice of the reviewer and made it clearer that both pseudo-experiments were derived from the same smog chamber experiment, but with a different number of volatility bins in each one of them.

**(16)** Fig. 1 – 3 and S1 - S3: The information about the content of each panel in these figures is there. But labelling the panels with a, b, c etc. in each figure will make it easier for the reader. Currently, only the Figure is referenced in the text and the reader has to figure out which of the panels is meant in the text.

We have labelled the panels in each one of these figures to avoid confusion.

**(17)** When the authors compare the different cases, e.g., when adding the additional high $C_{OA}$ data point, it is difficult to judge how much the reconstructed VBS distribution, yield curves, etc. really change from the base case. E.g., Fig 8a needs to be compared with the 25 °C panel in Fig 1 which has different x and y axis scaling. It would be very helpful to add the base case lines/bars to Fig 8 and 9. Especially the extrapolation of the base case to 200 µg m$^{-3}$ should provide an even stronger argument why the extra data point is useful.

We have followed the advice of the reviewer and added the base case line together with the case with the yield measurement at 200 µg m$^{-3}$.

**(18)** I need more information about the weighted averaging of the selected <5% solutions. When the averages are calculated, is each data point treated individually? I.e., data point 1 in solution 1 is close to the true data and gets a high weight. But data point 3 of solution 1 is far away (i.e., the slope of the solution is wrong). Does data

point 3 then get a different weight than data point 1? Or do all data points of a solution get the same weight factor?

For every combination of $\alpha_i$, $\Delta H_{vap}$, and $\alpha_m$ the algorithm calculates one overall *NMSE* following Eq. (10). Therefore, all data points for each solution get the same weight factor. The weighting factors are used for the averaging of the solutions. This clarification has been added to the paper.

**(19)** How many solutions usually are in the <5% group? Did this number differ between the investigated cases? E.g., were there less acceptable solutions when the "wrong" VBS range was chosen? If data points were treated individually, how much did the number of <5% solutions vary for the data points?

For Tests A1, A2, A3 and A4 the number of solutions under the <5% threshold were 148, 16, 16 and 115 respectively (out of 126,120 simulations). For Tests B1 and B2, the acceptable solutions were indeed fewer (82 and 50 respectively). For Tests C1 and C2, the acceptable solutions were 3,479 and 1,067 respectively. This information has been added to the manuscript.

**(20)** Following up on the previous comment assuming that each data point is treated individually: The authors could consider improving the visualisation of the range of estimates. Instead of a uniform grey area, they could indicate the density of solution curves with a colour scale. That would preserve the range of slopes of the solutions. Two examples of such figures are shown in the lower half of Fig. 2 in Li et al. (2019) and in Figure S1 from Tikkanen et al. (2019).

We have followed the recommendation of the reviewer and replaced the uniform grey area, with a shaded one showing the density of the solutions.

***Typos and language comments***

**(21)** Line 29: "The predicted yield uncertainty…" I find this sentence hard to understand.

We have rewritten this sentence to make it clearer.

**(22)** Line 35: "IPCC, 2013" should be updated to IPCC, 2021 unless the authors are referring to something very specific which is only contained in the 5th assessment report.

We have updated the IPCC reference.

**(23)** Line 103: comma before "respectively"

Added.

**(24)** Line 184: "SOA partitioning model" does that refer to the Eq. 1-3 in section 2.1?

Yes, the SOA partitioning model refers to Eq. (1) – (3). We have added the corresponding equation numbers at this point of the manuscript.

**(25)** Eq. 9: One sum goes up to "N0" the other to "n". Are these indeed different numbers or is it a typo?

This is a typo and has been corrected. Both refer to the total number of observations.

**(26)** Line 219: make it clear that these are the same models that are used by the analysis algorithm.

We have added the recommended clarification at this point.

**(27)** Line 365: "wider range" is not a precise description here. Better say "at higher SOA concentrations". Wider could also mean extending the range to lower concentrations.

We have replaced the statement "wider range" with "at higher SOA concentrations" following the suggestion of the reviewer.

**(28)** Line 304 "(20 to 200 °C with a step of 5 °C but including TD MFR values greater than zero)" I do not understand the "but…" part. If "but only including" was meant, how could there even be MFR values <0?

We have rephrased the sentence clarifying that we do not include zero values to avoid the division by zero.

**(29)** Line 306 "[…] we used a higher resolution for the first 0.5 h (step of 2 min), in which the evaporation is usually faster, and lower then (step of 10 min) up to 3 hours." What is meant with the "lower then …" part?

We assume that the sampling timestep is not constant in the isothermal dilution experiment. For the first 30 minutes there is a measurement every 2 min and after 30 min there is a measurement every 10 min. The sentence has been rephrased.

**(30)** Line 445 "[…] instead of the 4 bins used in Test A1 and covering the $10^3$ µg m$^{-3}$ material. "  I am not sure about the meaning of the underlined part in this sentence. This sentence has been rephrased.

**Responses to the Comments of Reviewer 3**

**(1)** Uruci et al. extended an algorithm by combining SOA yield measurements, thermodenuders (TD), and isothermal dilution (TD) to constrain the volatility product distribution ($\alpha_i$), effective vaporization enthalpy ($\Delta H_{vap}$), and effective accommodation coefficient ($\alpha_m$) and finally reduce the uncertainty of parameterization of SOA formation. The topic is worthy of investigation as the parameterization of SOA is a significant source of uncertainty in air quality modeling and is also suitable for publication in AMT. The overall writing is clear. However, there are some weaknesses in the result and discussion part that needed to be concerned prior to publication.

We would like to thank the reviewer for the constructive comments. Our responses and the corresponding changes in the manuscript (in black) follow each comment of the reviewer (in blue).

*Major comments*

**(2)** The main purpose of the study is to combine SOA yield measurements with TD and ID to reduce the uncertainties of estimation of $\alpha_i$, $\Delta H_{vap}$, and $\alpha_m$. However, there is no comparison of estimated parameters between derived from combining approach tests and derived from single set tests to indicate the improvement.

We have followed the advice of the reviewer and rerun the algorithm for tests A1, B1, and C1 without providing as input the corresponding SOA yield measurements. In this way the algorithm is practically the same as that of Karnezi et al. (2014). In all cases, the absence of the yield measurements led to a significant deterioration of the ability of the algorithm to estimate SOA yields at all temperatures and concentrations. For example, in test A1 the SOA yield error of the algorithm in the 5-35 °C temperature range increased from 14-24% to approximately 100%. The corresponding uncertainty range also increased by a factor of 4-6. Similar results were obtained in the other tests. We have added a new section in the revised paper where we compare the results of the new algorithm to those of the original and demonstrate the significant improvement in the ability to estimate SOA yields at different conditions.

We have also repeated selected tests using all combinations of the measurements (Yields/TD, Yields/Dilution and TD/Dilution). The Yield-TD combination gave the best results out of the three pairs. The isothermal dilution measurements are the least valuable because only a relatively small fraction of the SOA

evaporates and therefore the information provided is relatively limited and focuses on the more volatile components of the particles. Also, TD measurements are important to constrain well $\Delta H_{vap}$ and allow the more accurate extrapolation of the results to other temperatures. also provides information for the volatility distribution of the OA. The results of our tests (YE+TD, YE+DIL, TD+DIL) for Case A1 have been added to the Supplementary Information and their discussion to the main paper.

**(3)** The estimates of SOA yields, TD, and ID in all cases were good with $MNE_M$ of ~25% or less (Table 2), but the estimated $\alpha_m$ dramatically deviated from the truth value for almost all tests (Table 1). It indicates 'the difficulty of constraining $\alpha_m$.' Does this mean that the extended approach fails to improve the estimation of parameters, or at least for accommodation coefficients? In other words, why should we focus on the accommodation coefficients to decrease the uncertainty of parameterization of SOA yields?

Our results indicate that the effect of the mass accommodation coefficient on the measured quantities is relatively small compared to the other parameters (volatility distribution, enthalpy of evaporation) so it is difficult to constrain it. This conclusion is consistent with the results of Karnezi et al. (2021). The addition of the SOA yields to the inputs does not make much of a difference, because these are not affected by the accommodation coefficient. Other measurement approaches (e.g., evaporation experiments with monodisperse ultrafine particles) are needed to better constrain this quantity. A brief discussion of this point has been added.

**(4)** For the description of Experiment B1 in section 4.2.2, the saturation concentrations of LVOCs, SVOCs, and IVOCs range from $10^{-2}$ to $10^4$ μg m$^{-3}$, but the SOA yields of α-pinene (precursor for Experimental B) are quite different from them. I wonder whether it is applicable to use a larger volatility bin for this oxidation system, as Pathak et al. (2007b) reported that 'the performance of the both 4- and 7-product basis set parameterizations is similar for practically all data points.'

The objective of this test was to quantify the effect of using fewer volatility bins (a narrower volatility distribution) than that of the actual SOA system studied. This is often done in practice because the chemical transport models use a predefined relatively low number of volatility bins (e.g., 1, 10, 100, and 1000 μg m$^{-3}$) for all SOA systems. One could of course, use more volatility bins in the fitting with a corresponding increase

of the computational time. However, in this case the test would be quite similar to Experiment A and would add little to the analysis. An explanation of the reasons for the choice of a narrower volatility distribution has been added to the paper.

**(5)** Table 2 lists mean normalized errors of measurement data vs 'True' values ($MNE_T$) and vs estimated values ($MNE_M$). I wonder if the authors try to use similar values of $MNE_T$ and $MNE_M$ to support the good performance of the extended approach. And what is the reason for causing the small difference between them?

$MNE_T$ and $MNE_M$ were used to quantify the performance of the algorithm. Both $MNE_T$ and $MNE_M$ were quite close to the introduced experimental error. The difference was explained by both the "noise" introduced to the "measurements" that affects $MNE_M$ and the greater number of points used to calculate $MNE_T$. This is now clarified in the paper.

**(6)** What do 'close to unity' and 'resistances to mass transfer are small' mean in Line 370? The authors may want to include a description of the accommodation coefficient in the introduction section.

We have added a description of the accommodation coefficient and its effect on the results of evaporation experiments in the introduction and in the model description. There we explain that this parameter has been traditionally used to account for resistances to mass transfer not only at the surface of the particle but also inside the particle and also that the evaporation rate for most particles is relatively insensitive to its value when it is around one.

*Language suggestions*

**(7)** Line 20: add a comma between 'approach' and 'we'.
Corrected.

**(8)** Line 29: 'less' should be changed to 'smaller'.
Changed.

**(9)** Line 74: give the whole word when first using the abbreviation for 'TD', and also for 'LVOCs' in Line 112.
Corrected.

**(10)** Line 77: delete 'more'.

Deleted.

**(11)** Line 85-88: The sentence 'In TD applications in the …' is hard to understand, please rewrite it.

We have rewritten this rather confusing sentence.

**(12)** Line 100: delete 'two'.

Deleted.

**(13)** Line 101: 'type' should be changed to 'types'.

Done.

**(14)** Line 106: delete 'of'.

Deleted.

**(15)** Line 107: 'obtained' should be changed to 'conducted'.

Changed.

**(16)** Line 114: add a comma between 'enthalpy' and 'we'.

Added.

**(17)** Line 119: replace 'so' with 'thus' or 'therefore', and change throughout the text.

Changed throughout the text.

**(18)** Line 148: change the sentence 'The time-dependent evaporation of SOA in the TD is described in this work…' to 'The time-dependent evaporation of SOA in the TD used in this work is described…'

Revised.

**(19)** Line 288: delete 'also against'.

Deleted.

**(20)** Line 403: add a comma between 'problems' and 'the'.

Added.

**(21)** Line 457: Please simplify the sentence '…covering in that way…' to make it easy to understand.

This phrase has been rewritten.